# Domain Generalization for Time Series: Enhancing Drilling Regression Models for Stick-Slip Index Prediction

**Hana YAHIA**                                                    *hana.yahia@minesparis.psl.eu*
*Centre Automatique et Systèmes (CAS)*
*Mines Paris, PSL University*
*DrillScan, Helmerich & Payne*

**Bruno FIGLIUZZI**                                              *bruno.figliuzzi@minesparis.psl.eu*
*Center for Mathematical Morphology (CMM)*
*Mines Paris, PSL University*

**Florent DI MEGLIO**                                          *florent.di_meglio@minesparis.psl.eu*
*Centre Automatique et Systèmes (CAS)*
*Mines Paris, PSL University*

**Laurent GERBAUD**                                          *laurent.gerbaud@minesparis.psl.eu*
*Centre de Géosciences*
*Mines Paris, PSL University*

**Stephane MENAND**                                          *Stephane.Menand@hpinc.com*
*DrillScan, Helmerich & Payne*

**Mohamed MAHJOUB**                                        *Mohamed.Mahjoub@hpinc.com*
*DrillScan, Helmerich & Payne*

**Reviewed on OpenReview:** *https://openreview.net/forum?id=nNN1pPJRVL*

## Abstract

This paper provides a comprehensive comparison of domain generalization techniques applied to time series data within a drilling context, focusing on the prediction of a continuous Stick-Slip Index (SSI), a critical metric for assessing torsional downhole vibrations at the drill bit. The study aims to develop a robust regression model that can generalize across domains by training on 60 second labeled sequences of 1 Hz surface drilling data to predict the SSI. The model is tested in wells that are different from those used during training. To fine-tune the model architecture, a grid search approach is employed to optimize key hyperparameters. A comparative analysis of the Adversarial Domain Generalization (ADG), Invariant Risk Minimization (IRM) and baseline models is presented, along with an evaluation of the effectiveness of transfer learning (TL) in improving model performance. The ADG and IRM models achieve performance improvements of 10% and 8%, respectively, over the baseline model. Most importantly, severe events are detected 60% of the time, against 20% for the baseline model. Overall, the results indicate that both ADG and IRM models surpass the baseline, with the ADG model exhibiting a slight advantage over the IRM model. Additionally, applying TL to a pre-trained model further improves performance. Our findings demonstrate the potential of domain generalization approaches in drilling applications, with ADG emerging as the most effective approach.

# 1 Introduction

Despite recent advancements, developing a robust and generalizable machine learning (ML) model for predicting drilling malfunctions remains a significant challenge. A primary reason lies in the quality of data, which must be both large and diverse to encompass the variety of cases needed for effective model training. However, real world datasets often face issues such as bias, incompleteness, or insufficient labeling. Another significant challenge is dataset shift. Traditional machine learning models usually assume identical distributions for training and target data, an assumption that rarely holds in real world applications, complicating generalization further in fields like drilling.

During drilling, three types of vibrations can occur based on their direction: axial, lateral, and torsional. They can exist separately, which is rare, but usually they are synchronized in coupled modes. These vibrations can result from the bit-rock interaction or from the contact between the drill string and the borehole. In the present paper, we focus on torsional vibrations and more specifically on their most destructive type: the so-called stick-slip phenomenon. Stick-slip is among the most damaging types of vibrations that can affect the drill string, as it reduces rate of penetration (ROP), slows down the drilling process, and decreases efficiency (Zhu et al., 2014). This phenomenon results in periodic, irregular downhole rotation speed, where the bit rotational speed alternates between sticking (complete stop) and slipping (surpassing the surface rotation speed multiple times) phases (Shen et al., 2017). Stick-slip occurs when the rotational energy in the drill string is insufficient to overcome the torque on the bit (TOB). While drilling, the top drive continues to supply rotational energy to the drillstring, but a sticking phase can occur in its lower section, lasting between 1 to 5 s due to the high TOB. The accumulated torsional energy is then abruptly released, resulting in a sudden spike in bit velocity during the slip phase. Throughout the stick-slip vibrations, the surface rotation speed can remains constant.

Advanced downhole sensors allow for precise detection and measurement of stick-slip vibrations. However, these sensors can be costly, and real-time transmission of downhole data to the surface is not feasible with standard methods like mud telemetry. While faster data transmission is possible using wired drill pipes, it comes with significantly higher costs. Additionally, mud telemetry's bandwidth limitations restrict surface transmission to low-frequency data, meaning high-frequency data can only be accessed after drilling is completed. This is why we aim to detect stick-slip vibrations using surface measurements.

With advancements in machine learning models and data analysis techniques, many researchers have adopted data-driven approaches to automatically detect downhole events, thereby reducing the need for frequent crew inspections (Saadeldin et al., 2023; Elahifar and Hosseini, 2024). For instance, Zha and Pham (2018) developed a binary classification model utilizing surface data at 100 Hz to detect stick-slip, while Baumgartner and van Oort (2014) classified high-frequency downhole acceleration data at 400 Hz into stick-slip/no-stick-slip and whirl/no-whirl categories. Additionally, Hegde et al. (2019) compared machine learning algorithms, including logistic regression, support vector machines (SVM), and random forests, to classify stick-slip severity based on surface measurements. These last models were tailored for specific geological formations under the assumption that lithology, bottom hole assembly (BHA), drill bit, and drilling fluid remain constant. If any of these parameters change, new models must be trained using updated data to reflect those changes.

A common challenge with data-driven machine learning models for detecting downhole vibrations is their limited ability to generalize effectively. These models often perform well on wells within the same drilling field as the training data but experience significant drops in prediction accuracy when applied to wells outside this distribution, primarily due to differences in data distributions between training and test wells (Fang et al., 2020). Consequently, training a new model for each test well often yields better results.

To address this issue in drilling applications, we explored various data normalization methods to enhance model generalization, as well as the application of Transfer Learning (TL) techniques to develop a more robust and adaptable model for predicting the Stick-Slip Index (SSI) (Yahia et al., 2024a). Additionally, some researchers have incorporated physics-based principles into model training to improve generalization. For example, a Physics-Informed Machine Learning (PIML) approach (Sheth et al., 2022) utilized historical well logs and prior stand data to predict upcoming stick-slip classes, integrating a physics-based Stick-Slip Index (SSI) as a feature to boost classification accuracy. Similarly, we incorporated physical features

alongside surface measurements to develop a more generalizable model for predicting the SSI (Yahia et al., 2024b).

Beyond the drilling application, numerous methods have been proposed to enhance the generalization of machine learning models. Recent advances focus on aligning feature distributions between source and target domains through Domain Adaptation (DA) techniques, aiming to reduce domain discrepancies and improve target domain performance using existing source data. Some approaches achieve this by reweighting or selecting samples from the source domain (Borgwardt et al., 2006; Gong et al., 2013), while others transform the feature space to map the source distribution onto the target (Baktashmotlagh et al., 2013; Gopalan et al., 2011). A key factor in these methods is how the similarity between distributions is measured. One approach matches distribution means in a reproducing kernel Hilbert space (Huang et al., 2006), while Fernando et al. (Fernando et al., 2013) propose mapping the principal axes of the distributions, and Ganin (Ganin and Lempitsky, 2015) introduce Adversarial Domain Adaptation (ADA), which modifies feature representations rather than relying on reweighting or geometric transformation. This approach, which employs a deep and discriminately trained classifier to measure distribution separability (Singhal et al., 2023; Li et al., 2024; Fang et al., 2024), is extensively used across fields like task and text classification (Kim et al., 2017; Xu et al., 2019), image classification for crack detection (Weng et al., 2023), Medical Image Analysis (Kollias et al., 2024), sentiment analysis (Ganin and Lempitsky, 2015; Shen et al., 2018; Li et al., 2017; Xia et al., 2023) or Named Entity Recognition (NER) (Naik and Rose, 2020).

However, Domain Adaptation (DA) can only be applied when target domain data is available during training. In cases where this is not feasible, a more challenging and realistic approach, known as Domain Generalization (DG) (Matsuura and Harada, 2020), is preferred for practical applications. The primary objective of DG is to train a model using one or multiple different but related source domains so that it can generalize effectively to unseen target domains. While ADA was originally proposed for domain adaptation, the adoption of this reasoning has motivated adaptations of this approach for DG (Sicilia et al., 2023; Matsuura and Harada, 2020; Wang et al., 2022; Zhou et al., 2022). Domain generalization has been widely tested, particularly on image datasets for object recognition tasks (Albuquerque et al., 2019), Fault Diagnosis (Zhao and Shen, 2022; Li et al., 2020), anti-spoofing (Liu et al., 2022; Jia et al., 2020), depersonalized cross-subject vigilance estimation problem (Ma et al., 2019), and bearing fault identification (Zheng et al., 2019).

While these techniques have been extensively tested on image data, this paper focuses on testing and comparing their effectiveness on time series data. Specifically, we compare a baseline model with an Adversarial Domain Generalization (ADG) and an Invariant Risk Minimization(IRM) models to predict the severity of downhole torsional vibration (SSI) using 60 second sequences of 1 Hz surface drilling data as inputs within a real drilling context. The models are tested on separate test wells not used in training, and the ADG and IRM are applied to improve the baseline model predictions. Additionally, we investigate the benefits of Transfer Learning (TL) on trained models, and compare their performance post-TL.

The paper is organized as follows: Section 2 provides a brief overview of the rotary drilling system and stick-slip vibrations. Section 3 outlines the theoretical framework of the tested techniques. Section 4 describes the models architectures, while Section 5 focuses on the training process and the results. Finally, concluding remarks are listed in Section 6.

## 2 Description of Rotary Drilling System and Stick-slip Vibrations

### 2.1 Description of rotary drilling system

Rotary drilling is the most common method of drilling in oil & gas and geothermal applications. As depicted on Fig. 1, it consists in using a drilling device called bit, which usually destroys the rock in one of the two following ways: either by a shearing action (drag bits) or by indentation (roller-cones bits). The bit is linked to a rig located at the surface by a drill string comprising a series of pipes and a lower part called Bottom-Hole Assembly (BHA). The drill bit has to be rotated and pushed against the rock formation. The force, or Weight On Bit (WOB), required for the bit cutters to engage the formation is obtained from the weight of the drill string. The rotation is applied either by the use of a rotary table or a top drive at the surface, and with assistance of a downhole motor in some applications. Downhole motors are particularly

used for directional wells and when the bit technology requires higher rotary speeds that are not technically possible with only rotation from the surface. They help avoiding drill string twist offs by allowing most of the rotational torque to be concentrated near the bit instead of lost through torque and drag on the drill string (Anderson et al., 1990). In addition, drilling fluids and a circulation system are used to suspend and carry the cuttings and to cool down the drill bit and downhole equipment. Other equipment can be used to improve the drilling efficiency such as shock subs to damp the vibrations caused while drilling hard formation, or drilling jars that are used to free the drill string by delivering an impact load when a stuck pipe is faced (Richard, 2001).

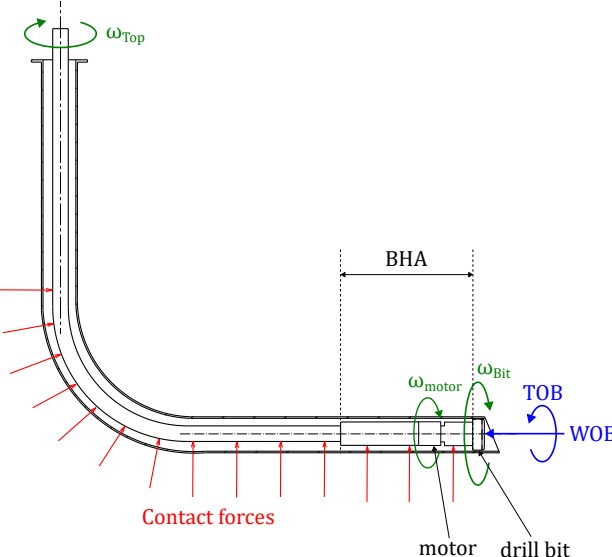

Figure 1: Simplified schematic representation of the drilling system

## 2.2 Drilling torsional vibrations: Stick-slip

Stick-slip is a periodic low-frequency torsional oscillation characterized by fluctuations in the bit rotation speed, transitioning from zero during the stick phase to several times the surface rotation speed during the slip phase. In cases where a downhole motor is utilized, it generally maintains a non-zero bit rotational velocity, however, severe stick-slip can still arise. To quantify the severity of stick-slip, we calculate the stick-slip index (SSI), an industry-standard metric that normalizes the fluctuations in bit rotation speed ($\omega_{Bit}$) over a specified time period, defined as follows:

$$SSI = \frac{\max \omega_{Bit} - \min \omega_{Bit}}{\overline{\omega_{Bit}}} \tag{1}$$

Fig. 2 depicts the changes in surface torque and downhole bit rotation speed over time throughout a severe stick-slip sequence. The bit experiences two distinct phases: the stick phase, during which the bit rotation speed is zero, and the slip phase, where the bit rotation speed reaches twice that of the surface rotation speed. While the surface rotation speed remains constant as imposed at the surface, the surface torque fluctuates in response to the variations in bit rotation speed.

Our objective is to train a generalizable regression model to detect stick-slip based on surface measurement. The model's inputs are 60 second sequences of surface features, including: surface torque, surface WOB, Rate Of Penetration (ROP), flow rate, and total rotation speed variations (surface rotation speed and downhole motor rotation speed), and the output is the predicted SSI. To ensure the model's generalizability to wells beyond those used in training, we explore the application of domain generalization techniques: ADG and IRM.

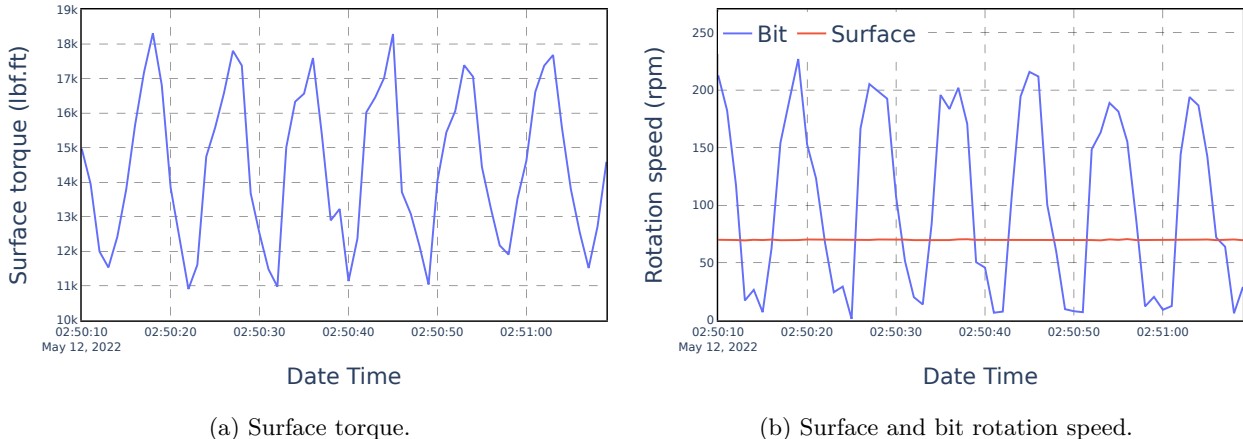

(a) Surface torque.

(b) Surface and bit rotation speed.

Figure 2: Example of a sequence with severe stick-slip.

## 3  Methodology

Generalization in machine learning refers to the capability of a model to perform effectively well on an unknown *target* domain that differs from the *source* domain on which it was originally trained. For drilling applications, we consider a training set comprising data from multiple wells, where each well is defined by distinct characteristics. Some are known, such as length, trajectory, and bit diameter, whereas others are unknown, uncertain or changing, such as rock characteristics or bit fatigue. The wells originate from different sites across various geographical locations and correspond therefore to distinct domains having their own unique data distribution. The goal is to train a model capable to achieve minimal prediction error when applied to test data from a new well, which constitutes a different domain with a data distribution not encountered during training. Unlike traditional machine learning approaches, which assume that training and testing data originate from the same distribution, the concept of model generalization focuses on achieving robust performance across various domains without prior knowledge of the target domain.

During training, we typically have access to labeled data from one or several source domains. For the target domain, we can encounter various situations depending on the nature of the available data:

- Only unlabeled samples from the target domains: this scenario is called unsupervised domain adaptation.

- Unlabeled samples from the target domains plus few labeled target samples: this scenario is referred to as semi-supervised domain adaptation.

- No samples from the target domain during training, meaning that the specific target domain in which the model will ultimately be deployed is completely unknown. This situation is called Domain Generalization (DG) and is the one considered in the present article. We present the two tested approaches of DG, ADG and IRM, in subsections 3.1 and 3.2, respectively.

When a small number of labeled target samples become available during testing, they can be utilized to further improve the performance of the trained model on the test data. This process, known as Transfer Learning (TL) (Zhuang et al., 2020), involves fine tuning the trained model to better fit the specific characteristics of the new dataset (target domain).

### 3.1 Adversarial Approach for Domain Generalization

Let us denote by $\mathcal{X}$ the instance set of measurements conducted on the wells. From a mathematical perspective, a *domain* is defined as a specific distribution $\mathcal{D}$ on the instance set $\mathcal{X}$. In our context, each well constitutes its own domain on $\mathcal{X}$. Suppose we have access to $N_S$ wells with labeled data, characterized by the *source domains* $\{\mathcal{D}_S^i\}_{i=1}^{N_S}$. Our objective is to construct a regression algorithm using data from the source domains that generalizes well to new domains.

To specify a learning problem, two ingredients are required: a distribution $\mathcal{D}$ on the instance set $\mathcal{X}$, and an unknown target function $f : \mathcal{X} \to \mathbb{R}$ that we seek to approximate. A classical approach for approximating $f$ is to map instances into a feature space $\mathcal{Z}$ using a representation function $\mathcal{R} : \mathcal{X} \to \mathcal{Z}$ and then select a function $h$ from a hypothesis class $\mathcal{H}$ that utilizes these features to perform the regression task. The representation function $\mathcal{R}$ induces a distribution on the feature space $\mathcal{Z}$, which we denote by $\tilde{\mathcal{D}}$. Furthermore, we denote by $\tilde{f} : \mathcal{Z} \to \mathbb{R}$ the function satisfying $f(x) = \tilde{f} \circ \mathcal{R}(x)$ for all $x \in \mathcal{X}$.

In our case, we dispose of $N_S$ labeled dataset $\{D_S^i\}_{i=1}^{N_S}$, where $D_S^i = \{(x_{S_i}^1, f(x_{S_i}^1)), \ldots, (x_{S_i}^{m_i}, f(x_{S_i}^{m_i}))\}$ consists of $m_i$ measurements sampled i.i.d from the source distribution $\mathcal{D}_S^i$. Our objective is to select a function $h$ from the hypothesis class $\mathcal{H}$, which minimizes the expected risk $\epsilon_T$ on the target domain, defined as:

$$\epsilon_T(h) = \mathbb{E}_{z \sim \tilde{\mathcal{D}}_T}[L(h(z), \tilde{f}(z))] = \mathbb{E}_{x \sim \mathcal{D}_T}[L(h \circ \mathcal{R}(x), f(x))], \tag{2}$$

where $L$ is the loss function used to predict the difference between the prediction $h(z)$ and the target value $\tilde{f}(z)$ and is usually selected to be the Mean-Squared Error (MSE), $D$ denotes the original distribution of the target data, while $\tilde{\mathcal{D}}_T$ represents the distribution of the target data transformed by the representation function $\mathcal{R}$. The main challenge here is that labeled data are not available for the target distribution, making the prediction of target error impossible to estimate directly. As a consequence, we need to rely on the available training data from the $N_S$ source domains and to adapt the training procedure to ensure good generalization.

Theoretically, it can be shown that to minimize the expected risk $\epsilon_T$ on a new domain using the available training source domains, one effective approach is to minimize the empirical risk on these source domains while simultaneously learning a representation that aligns the features across the source domains (Mansour et al., 2009; Albuquerque et al., 2019; Ben-David et al., 2006). Domain generalization algorithms aim to construct a shared representation for all source domains, which still ensures strong regression performance. In this article, we rely upon the adversarial approach presented in this paragraph to jointly learn a shared features representation for the source domains and predict the stick-slip index. These approaches to domain generalization are theoretically justified under the following assumptions (David et al., 2010):

- *Covariate shift*: the target function $f$ should remain consistent across all domains.

- *Similarity between features distributions accross domains*: the representation $\mathcal{R}$ should result in similar features for all source domains. To quantify this similarity, the *H-divergence* is commonly used (Ganin et al., 2016; Ben-David et al., 2006; 2010; Kifer et al., 2004). In practice, the H-divergence $d_h(\mathcal{D}_S^i, \mathcal{D}_S^j)$ between the source domains $\mathcal{D}_S^i$ and $\mathcal{D}_S^j$ is estimated by training a classifier $C$ to distinguish between domains $i$ and $j$ based on the features produced by the representation $\mathcal{R}$:

$$d_h(\mathcal{D}_S^i, \mathcal{D}_S^j) \simeq 1 - 2\text{err}_{\text{cl}}(C), \tag{3}$$

  where the classification error is estimated based on the empirical datasets $D_S^i$ and $D_S^j$ with respective sizes $m_i$ and $m_j$ as

$$\text{err}_{\text{cl}}(C) = \frac{1}{m_i + m_j} \sum_{k=1}^{m_i + m_j} L_{\text{cl}}\big(C(z_k), \mathbf{1}_{D_S^i}(z_k)\big). \tag{4}$$

  In Eq. (4) $L_{\text{cl}}$ is a classification loss, usually the cross-entropy loss and $\mathbf{1}_{D_S^i}(z)$ is the indicative function taking the value 1 when $z$ comes from dataset $D_S^i$. A high classification loss suggests that the classifier struggles to differentiate between the source domains. Conversely, a small H-divergence implies high similarity between the distributions $\mathcal{D}_s$ and $\mathcal{D}_t$.

- *Existence of a suitable mapping function*: There must exist a function $h^*$ that can accurately map features $z$ to their respective labels, regardless of the domain.

In our case, before applying DG to the regression model for SSI prediction using surface measurements, it is crucial to verify the validity of these assumptions. According to (David et al., 2010), only the last two assumptions are necessary for applying model generalization. Moreover, the covariate shift assumption is considered less restrictive, as it can be validated independently of the feasibility of model generalization. Regarding the second assumption, there is always a degree of similarity between feature distributions, even when the input feature distributions differ. This is because drilling operations typically rely on standardized tools and techniques, which help maintain consistency in surface measurement patterns across wells. Consequently, the overall structure of input feature distributions remains comparable across domains. For the last assumption, prior studies and data analysis have demonstrated a correlation between surface measurements, particularly surface torque, and the severity of stick-slip, supporting the assumption that a suitable mapping function exists. Furthermore, machine learning models have proven effective in capturing these relationships in similar applications, including detection of other downhole vibrations. For example, our work in (Yahia et al., 2024a) demonstrates the feasibility of developing a mapping function for SSI prediction. Even if the mapping function is not perfect, a regression model can still approximate it with a reasonable degree of accuracy.

Domain adversarial training is commonly used for learning domain-invariant features. This approach was originally introduced by Ganin and Lempitsky (Ganin and Lempitsky, 2015; Ganin et al., 2016) in the context of domain adaptation. As described in section 3.1, minimizing the target error $\epsilon_T$ on a new well amounts to reducing both the expected risks $\epsilon_{S_i}$ on each source domain and the discrepancy between source domains. To that end, we aim to learn a representation $\mathcal{R}$ that aligns the source domains but still produces features that remain relevant for the regression task at hand (Ganin and Lempitsky, 2015). To achieve this, adversarial training is used to learn domain-invariant features by simultaneously training two components: a generator and a discriminator in an adversarial manner. The input $x$ is processed by an embedding function or representation $\mathcal{R}$ (generator) to learn a domain invariant feature representation, which is used as input to the regression model $h$ for mapping to the stick-slip index. To ensure alignment between the source domains, a domain classifier $C$ (discriminator) is trained to distinguish between the source domains based on their embedded features. The goal is to maximize the domains classifier's loss, indicating that the embedded features from the source domains are sufficiently similar, making it difficult for the classifier to distinguish between them. For a dataset constituted of $m$ observations $\{(x_i, f(x_i)) \sim \mathcal{D}_S^{k_i}\}_{i=1}^m$, the final loss function is:

$$\min \frac{1}{m} \sum_{i=1}^m L\left(h(\mathcal{R}(x_i)), f(x_i)\right) - \lambda\ L_{cl}\left(C(\mathcal{R}(x_i)), \mathbf{1}_{D_S^{k_i}}(x_i)\right), \tag{5}$$

where the second term is used to penalize the distance between the embedded features for the source and target domains, $L_{cl}$ being the classification loss (cross-entropy loss) computed across the source domains, $\lambda$ being a weighting coefficient, and $\mathbf{1}_{D_S^{k_i}}$ is the indicative function that takes the value 1 if the element $x_i$ belongs to the source domain $D_S^{k_i}$, and 0 otherwise.

## 3.2 Invariant Risk Minimization (IRM)

Invariant Risk Minimization (IRM) is a DG technique aimed at learning a new data representation where the correlations remain invariant across different training environments, similar to ADG (Arjovsky et al., 2019). Specifically, the model learns a representation $\mathcal{R}(x)$, where $x$ represents the input features. The goal is for this representation to capture the underlying structure of the data invariant across the different environments. IRM seeks to identify a domain-invariant predictor $g = p \circ \mathcal{R}$ by searching for a regression function $p$ that is simultaneously optimal for all domains $\{D_S^i\}_{i=1}^{N_S}$, which lead to the optimization problem:

$$\hat{\mathcal{R}}, \hat{p} = \arg \min_{\substack{\mathcal{R}:\mathcal{X}\to\mathcal{Z} \\ p:\mathcal{Z}\to\mathbb{R}}} \sum_{i=1}^{N_S} \epsilon_{\mathcal{D}_S^i}(p \circ \mathcal{R})$$

$$\text{subject to } p \in \arg\min_{\bar{p}} \epsilon_{\mathcal{D}_S^i}(\bar{p} \circ \mathcal{R}), \forall i = 1, \ldots, N_S, \tag{6}$$

where $\epsilon_{\mathcal{D}_S^i}(p \circ \mathcal{R})$ is the empirical risk associated with the $i$-th domain. Eq. 6 is a challenging bi-level optimization problem. However, as demonstrated in (Arjovsky et al., 2019), when a linear classifier $p$ is fixed, any predictor $g = p \circ \mathcal{R}$ can be written as $g = p_*.\mathcal{R}_*$ for a scalar $p_*$. Without loss of generality, this scalar can be absorbed into $\mathcal{R}$, such that $\mathcal{R} = p_*.\mathcal{R}_*$, resulting in $g = 1.\mathcal{R}$. Thus, $\mathcal{R}$ itself becomes the invariant predictor. This reformulation simplifies the problem as follows:

$$\hat{\mathcal{R}} = \arg\min_{\mathcal{R}:\mathcal{X}\to\mathbb{R}} \sum_{i=1}^{N_S} \epsilon_{\mathcal{D}_S^i}(\mathcal{R}) + \alpha\|\nabla_{p|p=1.0}\epsilon_{\mathcal{D}_S^i}(p.\mathcal{R})\|^2, \tag{7}$$

where $\alpha$ is a penalty weighting coefficient used to balance the ERM term and invariance of the predictor $1 \cdot \mathcal{R}$ accross the different domains. We refer the reader to the original article (Arjovsky et al., 2019) for a detailed derivation of this result.

### 3.3 Transfer Learning

During the training phase, we do not have access to labeled target samples. However, during testing, some labeled samples may become available, which can be used to improve the performance of the trained model in this specific target domain. To achieve this, the trained model (source model) can be fine-tuned by adjusting all its parameters or selectively updating some of them using only the labeled target samples as retraining data (Zhuang et al., 2020). In our case, the source model is a regression model trained in various source domains. The fine-tuning process, using the small set of labeled target samples, is fast as a result of the limited amount of data involved compared to what is required to train a new model.

## 4 Models Characteristics

### 4.1 Baseline model

To determine the baseline model, we observed that the existing literature on predicting downhole vibrations remains relatively limited, with no clear consensus on the optimal architecture for addressing this problem. Recent studies have employed relatively straightforward machine learning techniques: for instance, (Saadeldin et al., 2023) utilized methods such as support vector machines (SVM), radial basis functions, and functional networks, while (Elahifar and Hosseini, 2024) relied on decision trees for stick-slip prediction. To the best of our knowledge, our approach (Yahia et al., 2024a) is among the first to leverage deep learning techniques for stick-slip index (SSI) prediction. We explored with various architectures, including Recurrent Neural Networks (RNNs), transformers, and Long Short-Term Memory networks (LSTMs). Ultimately, we selected the LSTM architecture, as it demonstrated superior performance in terms of both accuracy and execution time.

The baseline model architecture, based on Yahia et al. (2024a), consists of a single component that processes 60 second sequences of surface measurements as input and predicts the SSI. The architecture is built using a series of LSTM layers combined with Layer Normalization (LN) layers followed by an output dense layer with a single neuron to generate the SSI prediction. The loss function used is the MSE.

**Transformer model:** In recent years, the transformer model (Vaswani, 2017) has gained popularity over LSTM, as it eliminates the sequential processing of LSTMs by using self-attention mechanisms to process all time steps in parallel. This parallelization is often seen as advantageous for time series tasks. However, in choosing the architecture for the generator, we compared LSTMs architecture with transformers. During testing, the transformer model required significantly more time to train (training time increased by a factor

of five), and its performance was less favorable compared to LSTMs, which are more efficient with smaller datasets. For these reasons, we opted to use LSTM layers for the generator architecture.

### 4.2 Adversarial Domain Generalization model

For ADG, we adapt the global model architecture proposed by Ganin et al. (2016), which comprises three main components (see Fig. 3):

- **Generator (or feature extractor):** This component processes 60 second sequences of surface measurements, generating a D-dimensional embedded feature vector $z \in \mathbb{R}^D$. The generator parameters are indicated by $\theta_G$, and the relationship is expressed as $z = G(x; \theta_G)$, where $G$ is the same as $\mathcal{R}$ in Eq. (2). The generator architecture comprises a series of LSTM layers combined with LN layers, mirroring the baseline model except for the final output dense layer.

- **Discriminator (or domain classifier):** The discriminator takes the embedded features vector $z$ as input and classifies it based on annotated data from the source domain. Its parameters are represented by $\theta_C$, and its output is the predicted domain for each sequence: $C(z; \theta_C)$ (see Eq. (4)). The discriminator is structured as a fully connected neural network, with a Gradient Reversal Layer (GRL) as input layer to reverse the direction of the gradients during backpropagation (multiply the classification loss by $-\lambda$ (see Eq. 5)).

- **SSI-predictor:** This component utilizes the embedded features from the generator to predict the stick-slip severity index (SSI) for each sequence. The parameters for this mapping are denoted by $\theta_{SSI}$ and the regression output is $h(z; \theta_{SSI})$ (see Eq.(2)). The SSI-predictor is also a fully connected neural network.

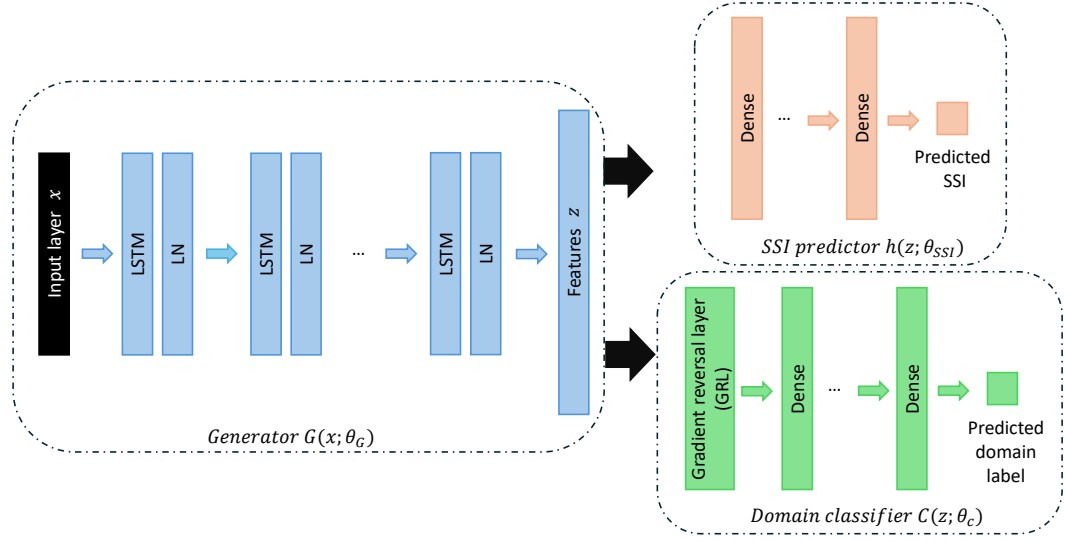

Figure 3: Architecture of ADG model. The generator is composed of a sequence of LSTM and LN layers; the SSI predictor and domain classifier are fully connected neural networks; a Gradient Reversal Layer (GRL) is used to reverse the direction of the gradients during backpropagation.

During the training phase, our objective is to minimize the SSI prediction loss for the training wells while optimizing the generator parameters to make the embedded features $z$ (where $z = G(x; \theta_G)$) domain-invariant, which involves aligning the source distributions $\{G(x_i, \theta_G), x_i \sim D_{s_i}\}$ for $i \in \{1, 2, ..., N_s\}$, where $N_s$ is the number of training wells. To achieve this, we aim to find the generator parameters $\theta_G$ that maximize the domain classifier loss, promoting domain-invariance and making the feature distributions across domains as

similar as possible. Simultaneously, we optimize the domain classifier and the SSI-predictor parameters, $\theta_C$ and $\theta_{SSI}$ respectively, to minimize their losses. The corresponding optimization problem is given by

$$\hat{\theta}_C, \hat{\theta}_{SSI}, \hat{\theta}_G = \mathrm{argmax}_{\theta_C} \mathrm{argmin}_{\theta_G, \theta_{SSI}} E(\theta_G, \theta_C, \theta_{SSI}) = L(\theta_G, \theta_{SSI}) - \lambda L_{cl}(\theta_G, \theta_C). \tag{8}$$

In Eq. (8), $L$ is the loss function used for the SSI prediction (MSE), while $L_{cl}$ is the classification loss (cross-entropy loss), computed across the $N_s$ source domains. The parameters $\theta_C$ of the domain classifier are optimized to minimize the domain classification loss (Eq. (9)), the parameters $\theta_{SSI}$ of the SSI-predictor are optimized to minimize the SSI prediction loss, while the feature mapping parameters $\theta_G$ are adjusted to both minimize the SSI prediction loss and maximize the domain classification loss (Eq. (10)). The parameter $\lambda$ is a weighting coefficient used to control the trade-off between the losses (Ganin and Lempitsky, 2015).

$$\hat{\theta}_C = \mathrm{argmax}_{\theta_C} E(\hat{\theta}_G, \theta_C, \hat{\theta}_{SSI}) \tag{9}$$

$$(\hat{\theta}_G, \hat{\theta}_{SSI}) = \mathrm{argmin}_{\theta_G, \theta_{SSI}} E(\theta_G, \hat{\theta}_C, \theta_{SSI}) \tag{10}$$

### 4.3   Invariant Risk Minimization model

The IRM model adopts the same architecture as the ADG model, with the primary difference being the exclusion of the domain classifier component. Specifically, the IRM consists of the generator $G$, made up of a series of LSTM and LN layers, which processes 60 second sequences of surface measurements as input. The output from the generator is then fed into the SSI predictor $h$ (same as ADG), a fully connected neural network, which is responsible for predicting the Stick-Slip Index (SSI) (see Fig. 3). An additional trainable variable $\beta$ is introduced to represent the linear classifier $p$ in Eq. 11 and to compute the regularization term. So that the IRM loss function is:

$$\sum_{i=1}^{N_S} L(h(G(x_{\mathcal{D}_S^i}, \hat{\theta}_G), \hat{\theta}_{SSI}), f(x_{\mathcal{D}_S^i})) + \alpha \|\nabla_{\beta|\beta=1.0} L(\beta.h(G(x_{\mathcal{D}_S^i}, \hat{\theta}_G), \hat{\theta}_{SSI}), f(x_{\mathcal{D}_S^i}))\|^2, \tag{11}$$

Where $L$ is the MSE loss, $x_{\mathcal{D}_S^i}$ is the data from the $i$-th source domain, $f(x_{\mathcal{D}_S^i})$ represent the true SSI of $x_{\mathcal{D}_S^i}$, and $\alpha$ is a penalty weighting coefficient.

## 5   Experiments

In this section, we present a variety of results for training a machine learning model to predict the SSI using 60 second sequences of surface measurements as input. The model is trained using different training wells (source domains) and tested on completely different wells (target domain) in four different approaches:

- **Baseline model**: A traditional deep learning model consisting of a single component that takes the 60 second sequences of surface measurements as input and predicts the SSI as output.

- **ADG model**: As explained in subsection 4.2, the model consists of three components: a generator, SSI-predictor and domain classifier.

- **IRM model**: This generalization technique incorporates a regularization term into the loss function. It processes 60 second sequences of surface measurements as input for the generator, and the output is then used as input to the SSI predictor. The architecture is illustrated in subsection 4.3.

- **TL**: In contrast to the previous techniques, TL is applied during the testing phase. Once a model is trained (either the ADG, IRM, or baseline model), if labeled surface data sequences from the test well are available during testing, we can apply TL technique to improve the model's performance.

### 5.1 Data processing

Extensive evaluation are performed of the proposed approaches on a number of distinct wells as outlined in Table 1. Some of these wells are from the same field and share similar characteristics. To assess the generalizability of the trained model, we ensure that wells from the same field are all consistently used either as test data or training data, but not both. The first three wells, which originate from the same field and represent a substantial number of sequences, were selected for the training process. The last three wells were reserved for testing, as two of them are from the same field, with an additional distinct well included to further evaluate the model's generalizability. The remaining wells (4, 5, and 6) were used alternatively for training and validation (see section 5.2).

Table 1: Source and target well characteristics, where vertical wells are drilled straight down from the surface into the target formation or reservoir, and lateral wells are drilled vertically to a certain depth, then deviates horizontally within the target.

| Well | Field number | Type of wellbore trajectory | Well length (ft) | Bit diameter (inch) | Drill Pipe diameter (inch) | BHA length (ft) | Number of sequences |
|---|---|---|---|---|---|---|---|
| **1** | 1 | Lateral | 18 120 | 8 3/4 | 5 | 167.94 | 33 303 |
| **2** | 1 | Lateral | 18 204 | 8 1/2 | 5 | 166.9 | 50 285 |
| **3** | 1 | Lateral | 17 921 | 8 3/4 | 5 | 136.94 | 15 147 |
| **4** | 2 | lateral | 27 723 | 8 3/4 | 5 1/2 | 101.03 | 52 008 |
| **5** | 3 | Lateral | 19 355 | 6 3/4 | 5 1/2 | 128.99 | 119 613 |
| **6** | 4 | Lateral | 29 077 | 8 3/4 | 5 | 130.52 | 255 687 |
| **7** | 5 | Lateral | 28 645 | 8 3/4 | 5 1/2 | 97.36 | 449 002 |
| **8** | 6 | Vertical | 20 619 | 8 3/4 | 5 | 110.31 | 22 054 |
| **9** | 6 | Vertical | 20 679 | 8 3/4 | 5 | 111.23 | 136 824 |

For each well described in Table 1, 1 Hz time series of surface and downhole drilling data are collected. The surface data, which includes surface torque, surface weight on bit, rate of penetration, flow rate, and total rotation speed variations, is divided into 60 second sequences where each sequence serves as an input sample for the regression model. The downhole data contains the bit rotation speed used to label the surface data. For each 60 secondsequence, we inspect the downhole bit rotation speed and we calculate the true SSI as described in Eq. (1).

### 5.2 Grid search for hyperparameter tuning

To tune our model's hyperparameters, we employ the traditional grid search method with the ADG model. This approach involves exhaustively searching through a specified subset of the hyperparameter space for the training algorithm. The hyperparameters evaluated include the regularization parameter for the generator (covering bias, kernel, and recurrent regularization), the number of hidden layers in the generator, and the loss function weighting coefficient $\lambda$ for the ADG (Eq. (5)). The potential values for the regularization coefficient are $(10^{-3}, 10^{-4}, 10^{-5})$, the number of hidden layers in the generator are set to $(4, 6, 8)$, while the weighting coefficient $\lambda$ varies among $(1, 10, 100, 1000)$. In our experiments, validation data is used to determine the optimal hyperparameter values. Since our goal is to develop a model that generalizes effectively accross sources, we select validation data from wells entirely distinct from those used in training. Therefore, in each training session, from the first six wells listed in Table 1, four wells are used for training and the remaining two for validation. The last three wells (from two different rigs) are reserved for testing after the hyperparameters have been selected. Selecting the two wells for validation among the first six wells is challenging, as this choice significantly influences both model performance and hyperparameters selection: when wells with characteristics similar to the majority of training wells were chosen, the SSI validation error tended to be low. In contrast, choosing wells with different characteristics resulted in a higher validation error. To address this, we test three different validation data cases, as outlined in Table 2. In each case, the

first three wells in Table 1, which are from the same field, are fixed as training data, and wells 4, 5, and 6, from 3 different rigs, are used alternately to select two wells for validation in each scenario.

Table 2: Validation data selection for the three tested cases.

|  | Well 4 | Well 5 | Well 6 |
|---|---|---|---|
| **Case 1** | **Validation data** | Training data | **Validation data** |
| **Case 2** | **Validation data** | **Validation data** | Training data |
| **Case 3** | Training data | **Validation data** | **Validation data** |

### 5.2.1 Regularization coefficient

Due to limited computational capacity, we do not search for all the hyperparameters simultaneously. Instead, we begin by fixing the number of hidden layers in the generator at 6 while varying the regularization coefficient and the weighting coefficient for the ADG across the different validation data cases described in Table 2. The architectures for the domain classifier and SSI predictor were fixed at 5 dense layers, each containing $[60, 40, 20, 10]$ neurons for the first four layers, and 6 neurons (representing the number of training and validation wells) and 1 neuron (for SSI) for the last layers, respectively. The generator architecture comprised 5 LSTM layers combined with 5 layer normalization (LN) layers (including 2 input layers, 6 hidden layers, and 2 output layers), with 64 neurons per layer and bias, kernel, and recurrent regularization applied uniformly across all layers. Each architecture search was conducted for 500 epochs, utilizing a constant learning rate of $10^{-3}$ on an Nvidia GPU. For each combination of hyperparameters, and for each validation data case, the model was trained using three different initializations. For each initialization, the trained model was used to calculate the MSE and dynamic time warping (DTW) (Li, 2021) which is a computational technique used to measure the similarity between two time series that may vary in speed or timing, between the actual and predicted SSI validation data. The average of the three initializations was then computed. The key advantages of DTW are its robustness to outliers, making it less sensitive to abnormal points, and its ability to stretch or compress parts of the time series to achieve optimal alignment. In contrast, MSE calculates point-by-point differences in magnitude between the two time series, without accommodating variations in timing or alignment. However, MSE is generally easier to interpret, as it measures the average of the squared differences between predicted and actual values, making it more straightforward than DTW. To facilitate comparisons, we decide to normalize the DTW by the number of corresponding well sequences.

Fig. 4 illustrates the average MSE on validation data across the three tested initializations for each of the three cases. As we can see, results vary across cases, with case 2 exhibiting the lowest MSE SSI validation error. This outcome is likely due to the training dataset's sequence configuration, as for case 2, wells 4 and 5 were selected for validation, while well 6, with the highest sequence number, was included in the training dataset. Fig 5 displays the mean MSE and normalized DTW on validation data, calculated over the three tested cases. Notably, the error variations remain consistent whether using MSE or DTW. The results indicate that the lowest validation SSI prediction error is attained with a regularization coefficient of $10^{-4}$ for most of the tested weighting coefficients. Therefore, this hyperparameter will be set at $10^{-4}$ for the remainder of the paper.

### 5.2.2 Number of generator hidden layers and weighting coefficient

To select the optimal number of hidden layers for the generator and the weighting coefficient $\lambda$ (see Eq. 8), we conduct a grid search using three different initializations across the three validation cases (see Table .2), following the same approach as for the regularization coefficient. The tested values for the generator's hidden layers are $(4, 6, 8)$ and for the weighting coefficient $\lambda$ we set $(1, 10, 100, 1000)$. The optimal parameters are chosen based on the SSI validation error. Similar to the regularization coefficient selection, Fig 6 shows the average MSE on validation data across the three initializations for each case, while Fig 7 presents the mean MSE and normalized DTW on validation data, averaged across the three cases. We choose the hyperparameter configuration with the lowest SSI validation error, which includes 6 hidden layers for the

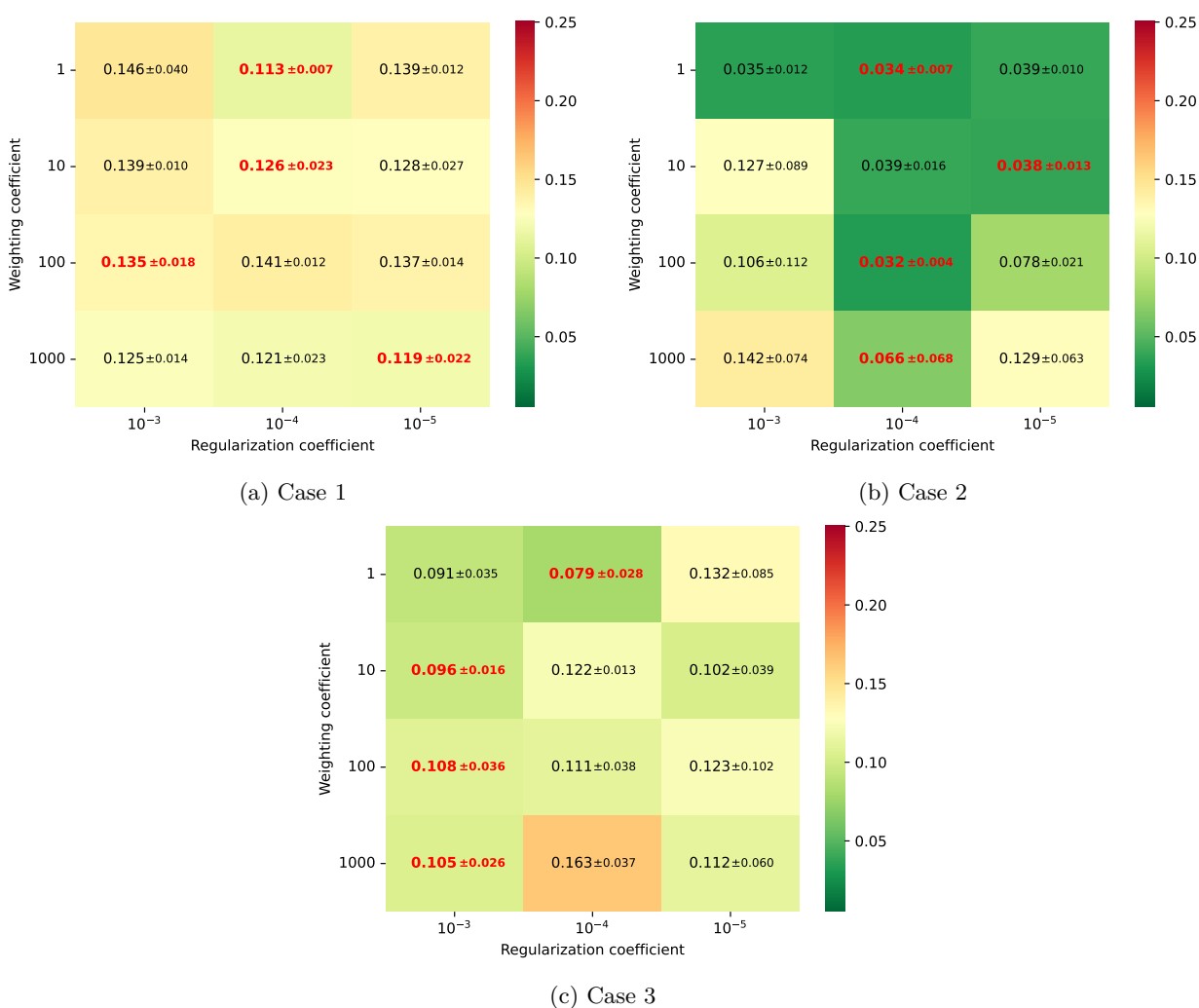

Figure 4: Average MSE of SSI validation data predictions for the three tested cases (Table. 2) with varying regularization and weighting coefficient over three different intializations: The ADG model is trained for each combination of hyperparameters using three distinct initializations. The validation SSI error is calculated for each one, and the average error is then computed across the three initializations.

generator and a weighting coefficient of 10. These hyperparameters will be fixed for the remainder of the paper.

**Remark:** For the penalty weighting coefficient $\alpha$ of the IRM model, we tested five different values [0.01,0.1,1,10,1000] and selected the model with the lowest SSI prediction loss, which occurred with a penalty coefficient of 1. This IRM model is then used for comparison with the ADG and baseline models for the rest of the paper.

## 5.3 Results and discussions

### 5.3.1 Comparison of ADG, IRM, and Baseline Models

With the selected hyperparameters, we proceed to compare the ADG and IRM techniques applied to time series for SSI prediction against the baseline model (section 4.1). For training, the first six wells are designated as training data and the last three as test data (Table 2), with 10% of the training data reserved for validation. Each model is trained for 1000 epochs, with a batch size of 2048, and a fixed learning rate of $10^{-3}$, and

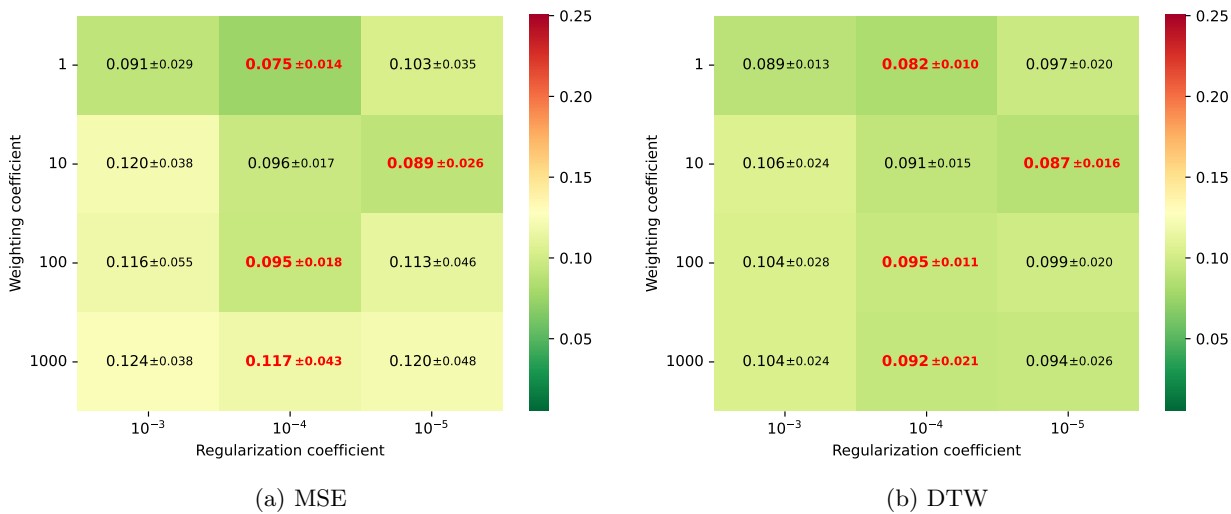

Figure 5: Average MSE and DTW of SSI validation data predictions for the three tested cases with varying regularization and weighting coefficient in the ADG model.

we retain the model that achieves the lowest SSI validation MSE. We repeat the training five times, each time with a different initialization, and we calculate the average test SSI prediction error across the five runs. The ADG model requires about six hours of training, whereas the IRM and baseline models each take approximately four hours.

Table. 3 shows the normalized DTW results for the three test wells, over the five runs, for the ADG, IRM and baseline models. The ADG and IRM models demonstrate better generalization compared to the baseline, achieving improvements of 10.86 % and 8.42 %, respectively. However, the ADG shows slightly better results than the IRM. Fig. 9 and Fig. 10 illustrate examples of true and predicted SSI sequences for the three tested models across the three test wells. As shown in Fig. 9a and Fig. 9b, all models generally perform well in predicting SSI for most sequences. However, there are also sequences where they all fail to capture SSI variations, such as in Fig. 9c, where they overestimate the SSI, and Fig. 9d, where they underestimate the SSI. Notably, in some cases (Fig. 10), the ADG and IRM provide more accurate SSI predictions than the baseline. This indicates that the generator effectively projects the training data into a domain-invariant feature space, whether with ADG or IRM, enabling the SSI predictor to make more reliable predictions using the newly projected features. Fig. 10b and Fig. 10c show sequences where the ADG model predicts the SSI with slightly greater accuracy than the IRM model, with both models consistently outperforming the baseline.

To further evaluate the results, Fig. 8 presents the confusion matrices for the three tested models, averaged over the three test wells. The SSI is classified into four distinct categories based on the severity of stick-slip: sequences without stick-slip (SSI $\in [0, 0.3[$) are assigned to the first class; sequences with low stick-slip (SSI $\in [0.3, 0.5[$) fall into the second class; sequences with moderate stick-slip (SSI $\in [0.5, 0.7[$) are grouped into the third class; and sequences with severe stick-slip (SSI $\geq 0.7$) are classified into the fourth class. As shown, the confusion matrices reveal that all models correctly identify the majority of sequences for each class, except for the last class, which corresponds to sequences with severe stick-slip, the most damaging scenario. The baseline model incorrectly assigns most of these sequences to the third class instead of the fourth, a misclassification not seen in the ADG and IRM models. The IRM and ADG models show similar performance, with the IRM model predicting SSI more accurately for sequences with low SSI, while the ADG model performs better for sequences with high SSI. The reasons for sequence misprediction are discussed in the subsection 5.3.3. In conclusion, the IRM and ADG models exhibit similar performance, with a slight advantage for the ADG, but both outperform the baseline model.

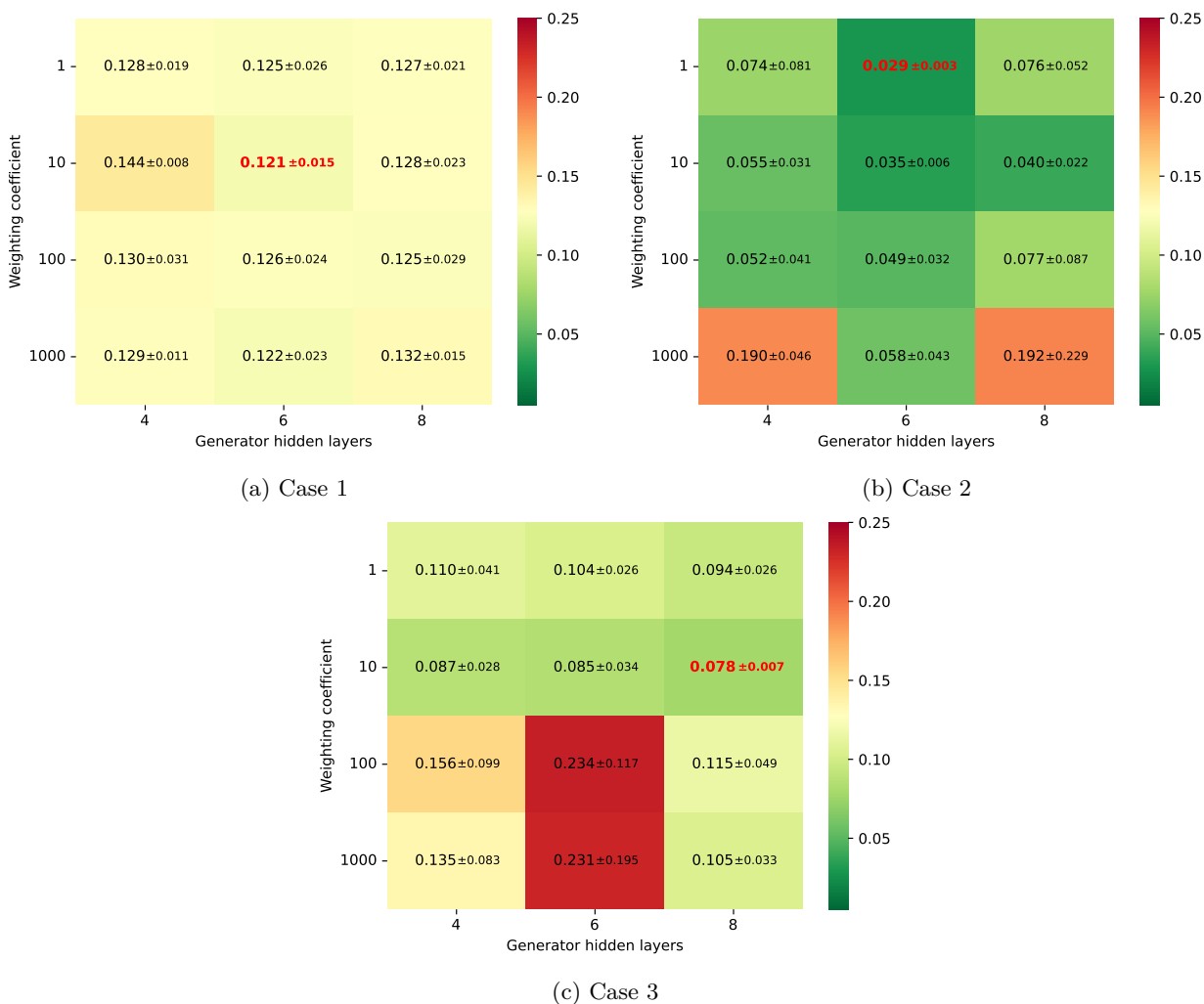

(a) Case 1

(b) Case 2

(c) Case 3

Figure 6: Average MSE of SSI validation data for the three tested cases with varying generator hidden layer number and weighting coefficient over three different intializations: The ADG model is trained for each combination of hyperparameters using three distinct initializations. The validation SSI error is calculated for each one, and the average error is then computed across the three initializations.

Table 3: Normalized SSI DTW of the trained ADG, IRMand baseline models on the three test wells: Each model is trained with five different initializations, and the error is reported as the average across these five runs.

|  | Baseline | ADG | ADG vs Baseline (%) | IRM | IRM vs Baseline (%) | ADG vs IRM (%) |
|---|---|---|---|---|---|---|
| Test well 1 | 0.136 | 0.122 | **10.29** | 0.120 | **11.76** | **-1.66** |
| Test well 2 | 0.086 | 0.075 | **12.79** | 0.080 | **6.97** | **6.25** |
| Test well 3 | 0.107 | 0.097 | **9.34** | 0.100 | **6.54** | **3.00** |

### 5.3.2 Transfer learning application

To enhance model performance on a specific target well, a TL technique can be applied if a small amount of labeled surface drilling data for that well is available (see section. 3.3). In our case, we test the contribution of TL on each of the three test wells by using the first 10 % of labeled surface drilling data with both, the

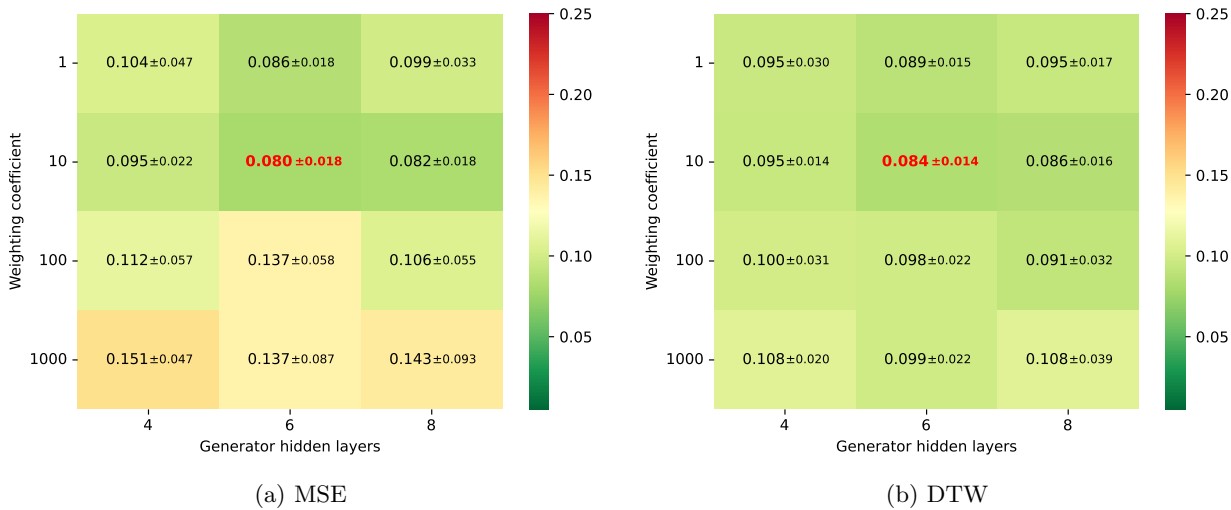

(a) MSE

(b) DTW

Figure 7: Average of the MSE and DTW of SSI validation data for the three tested cases with varying generator hidden layer number and weighting coefficient for the ADG model.

ADG and baseline models. The source model, either the trained ADG or baseline model, is then fine-tuned to adapt to the target well by retraining only the weights and biases of the first two layers in the generator and SSI predictor for the ADG model, and the first two layers in the baseline model. The same optimizer and learning rate as in the source model training are used, but with a significantly reduced number of epochs, resulting in a lighter computational load, with a retraining time of approximately 1.5 minutes.

Table. 4 presents a comparison of model performance before and after applying TL for both ADG and baseline models. For each test well, the source model (either ADG or baseline) is partially retrained using the first 10 % of labeled surface drilling data. The results in the table show that TL enhances model performance in both ADG and baseline models, with a decrease in normalized DTW following retraining. Additionally, this TL approach is efficient in terms of time and requires only a small amount of data compared to what would be needed to train a new model.

Table 4: Comparison of model performance, for both ADG and baseline models, before and after applying TL, using normalized SSI DTW as the evaluation metric across the three test wells: Each source model undergoes partial retraining with the first 10 % of labeled surface drilling data from each test well.

|  | ADG | | | Baseline | | |
|---|---|---|---|---|---|---|
|  | **Without TL** | **With TL** | **%** | **Without TL** | **With TL** | **%** |
| **Test well 1** | 0.122 | 0.095 | **22.13** | 0.136 | 0.110 | **19.11** |
| **Test well 2** | 0.075 | 0.058 | **22.66** | 0.086 | 0.06 | **30.23** |
| **Test well 3** | 0.097 | 0.088 | **9.27** | 0.107 | 0.100 | **6.54** |

Fig. 11 shows the actual and the predicted values of SSI over time for the three test wells, comparing results with and without the application of TL. As we can see, after TL was applied, the new model predicts the SSI better than the source one for both models. Additionally, even after TL application, the ADG model outperforms the baseline model, with a lower normalized DTW across all three test wells (see Table. 4).

### 5.3.3 Causes of misprediction of SSI

The three models, baseline, ADG and IRM, before or after the application of TL, mispredict the SSI for certain sequences. In this section, we analyze some of the potential reasons for these mispredictions, which cannot generally be addressed by domain generalization approaches.

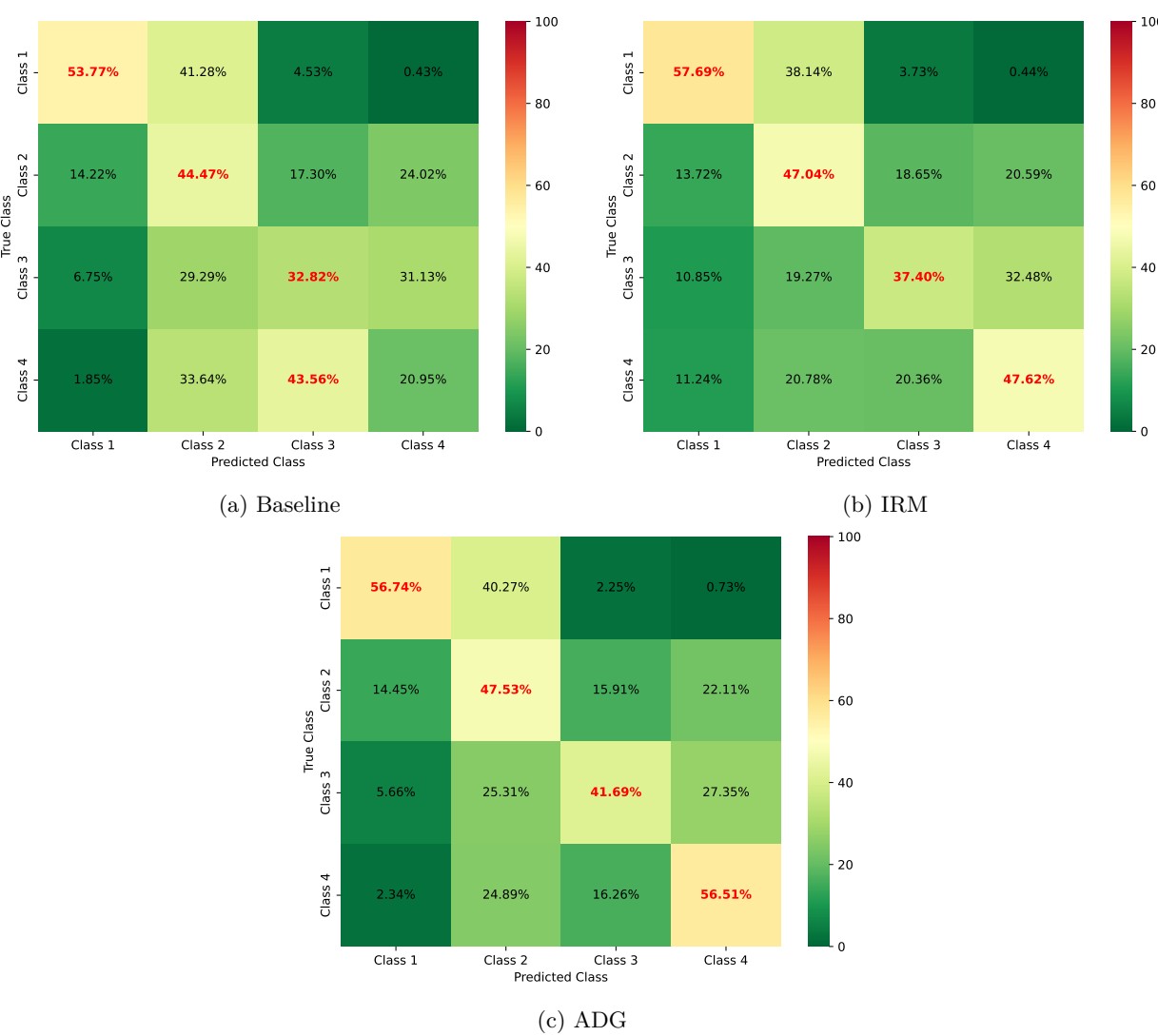

(a) Baseline
(b) IRM

(c) ADG

Figure 8: Confusion matrices illustrating the performance evaluation of the baseline, IRM and ADG models over the three test wells. The true and predicted SSI values are classified into four categories: sequences with $SSI \in [0, 0.3[$ are assigned to the first class; $SSI \in [0.3, 0.5[$ are classified into the second class; if the SSI $\in [0.5, 0.7[$ belong to the third class; and sequences with $SSI \geq 0.7$ are assigned to the fourth class.

**Synchronization problem**  In drilling operations, a time delay, commonly referred to as jet lag, occurs between surface and downhole measurements. This delay corresponds to the propagation time required for downhole data to reach the surface, and it varies over time. Consequently, achieving perfect synchronization between surface and downhole measurements is especially challenging, particularly during long-duration drilling.

To evaluate our trained models, we predict the SSI using surface measurements and compare these predictions with the true SSI, which is calculated using synchronized downhole measurements (Eq. 1). However, due to imperfections in synchronization, discrepancies can occur, leading to errors in evaluation, as illustrated in the following example. The synchronization issue is present in most domains and cannot therefore be addressed with domain generalization techniques.

Fig. 12 shows a long test sequence that exhibits synchronization issues between surface and downhole data, with the surface data being significantly shifted. In this sequence, we used the 60 second red and blue

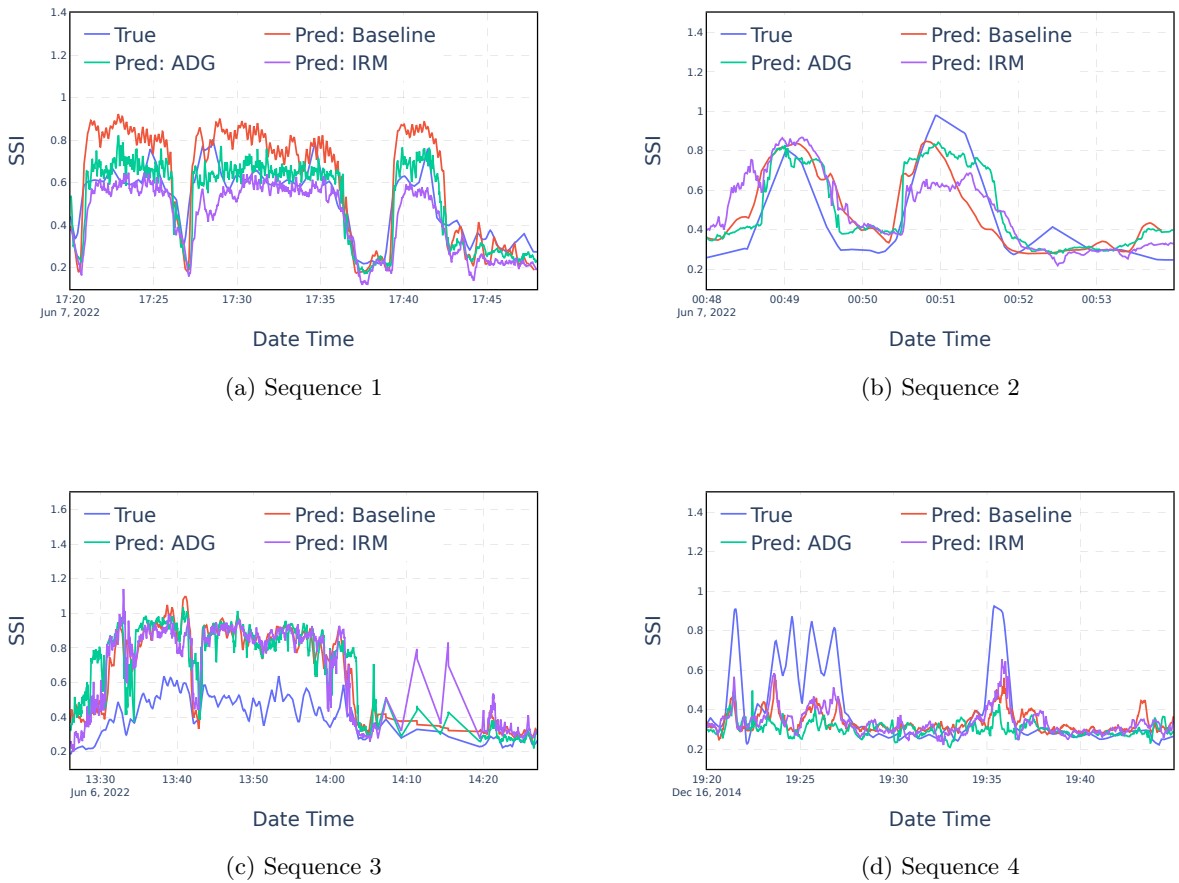

Figure 9: True and predicted SSI values over time for the three trained models: baseline, IRM, and ADG, based on sequences from the three test wells. In the first two sequences, all models accurately predict the SSI. In the last two sequences, all models fail to capture the variations in SSI.

sequences to predict their SSI using the trained ADG model. The model mispredicts the SSI of the red sequence, where the predicted SSI is high (0.77), due to surface torque variations, while the true SSI, calculated from the mismatched downhole measurements, is much lower (0.1) due to the absence of downhole vibrations. For the blue sequence, the predicted SSI is 0.84 and the true SSI is 0.92, indicating that the model predicts the SSI fairly well. This is because surface torque variations are observed, likely caused by downhole vibrations, even though they do not correspond to the true sequence.

**Unobservable stick slip** In some instances, downhole stick-slip can occur without generating vibrations detectable at the surface. This suggests that the downhole vibrations were attenuated as they propagated to the surface, due to friction and viscous damping, making them undetectable with low-frequency (1 Hz) surface measurements. This phenomenon is more commonly observed in lateral wells, particularly within the horizontal section. Fig. 13 illustrates the variation in surface torque and bit rotation speed during a test sequence in the horizontal section of a Test well. As shown, severe stick-slip occurs at the bit (with a true SSI of 0.87), but no surface torque variation is observed. Consequently, the trained model fails to predict the SSI using 1 Hz surface data for this type of sequence (with a predicted SSI of 0.14).

**Mislabeled data** To label surface data sequences, we compute the true SSI using downhole bit rotation speed measurements (Eq.1). However, some sequences may exhibit sudden variations (peaks) that do not

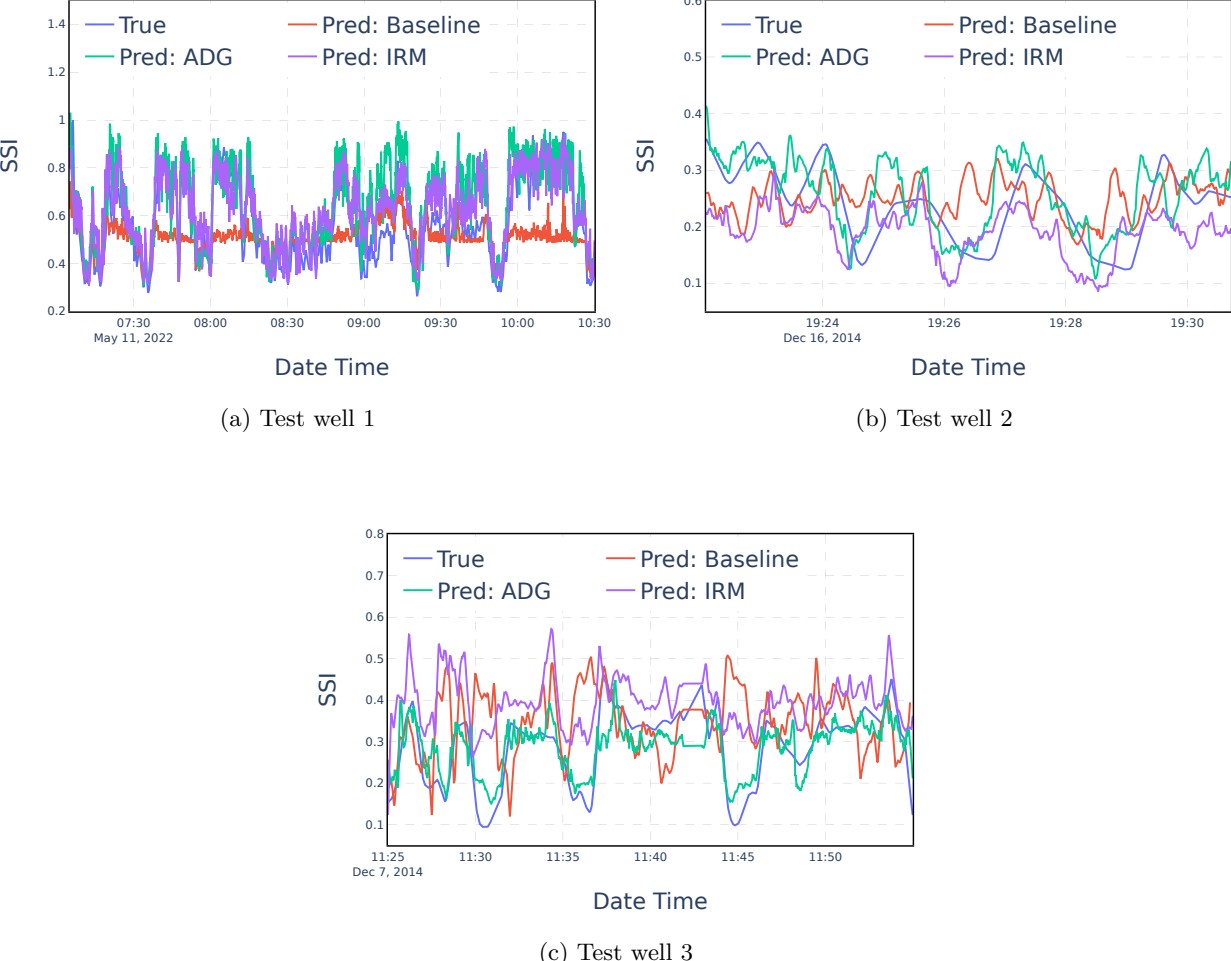

(a) Test well 1                    (b) Test well 2

(c) Test well 3

Figure 10: True and predicted SSI values over time for the three trained models: baseline, IRM, and ADG, based on sequences from the three test wells. For the three test sequences, the ADG and IRM provide more accurate SSI predictions than the baseline. The last two sequences the ADG model predicts the SSI with slightly greater accuracy than the IRM model.

signal stick-slip occurrences, but still result in high SSI values. Fig. 14 illustrates a test sequence where a sudden variation in downhole bit rotation speed results in a high true SSI (0.72). However, no corresponding changes are observed at the surface, causing the predicted SSI to be significantly low (0.21). This issue arises not from the trained model, but from the sequence labeling process.

**Domain mismatch** Another potential cause of ADG or IRM issues is domain mismatch, which arises when the model is not trained on enough data that adequately represents the target domain. Consequently, the model has difficulty generalizing effectively. To enhance generalization, the model may need to be trained on a more diverse dataset, which would improve its performance when exposed to new, unseen data.

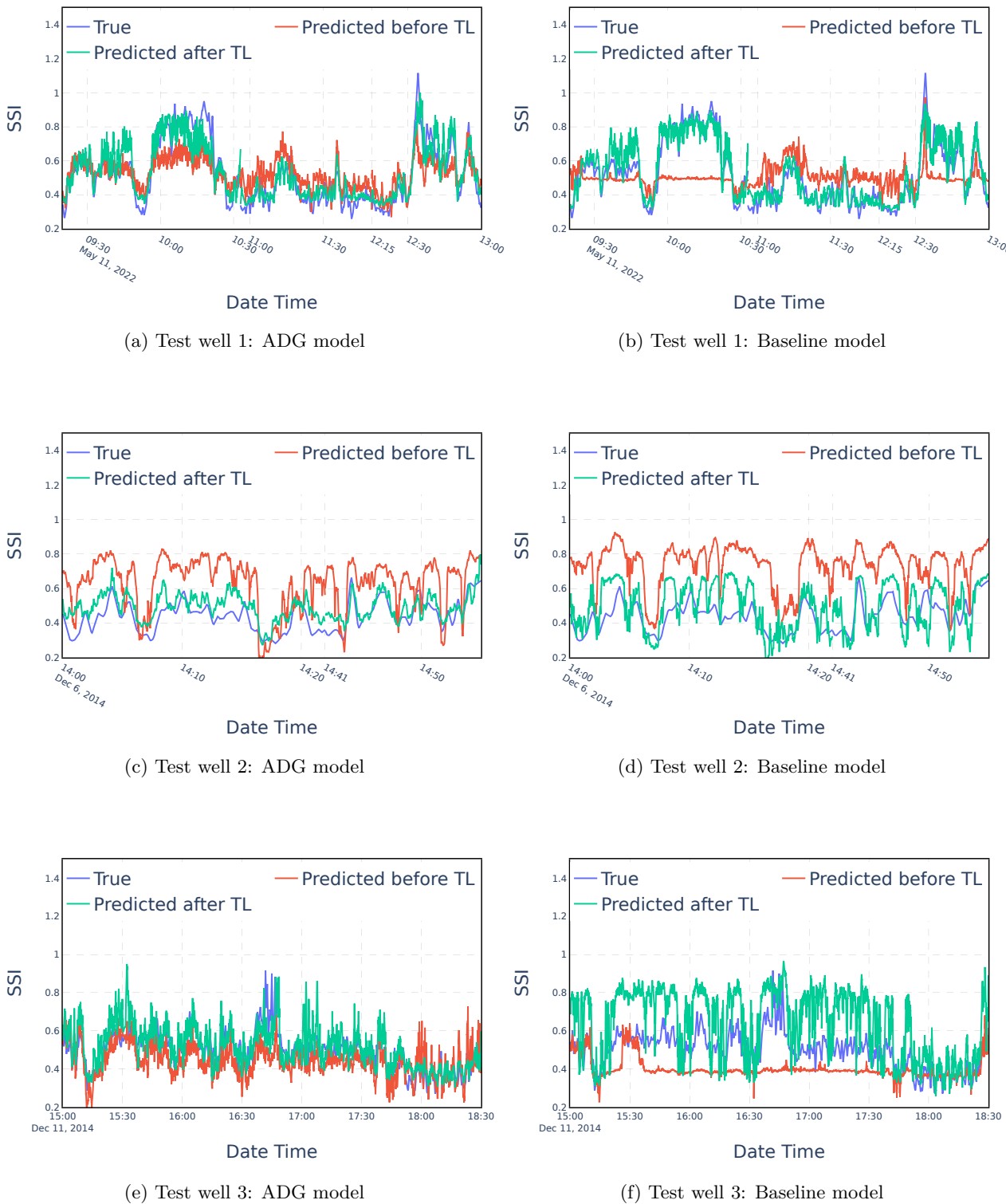

Figure 11: True and predicted SSI values over time, for both ADG and baseline models applied on sequences from the three test wells before and after the application of TL.

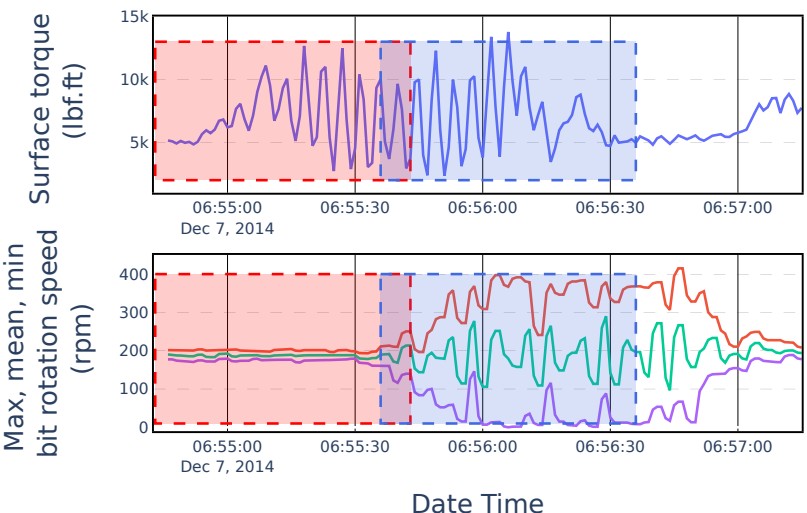

Figure 12: Variation in surface torque (surface measurement) and bit rotation speed (downhole measurements) for a test sequence. This sequence exhibits a synchronization issue between surface and downhole measurements. The predicted SSI for the 60 second red and blue sequences are high (0.77 and 0.84 respectively) due to torque variations. However, when using the mismatched sequence, the true SSI values for the red and blue sequences are 0.1 and 0.92 respectively. Consequently, the model mispredicts the SSI of the red sequence due to the synchronization problem.

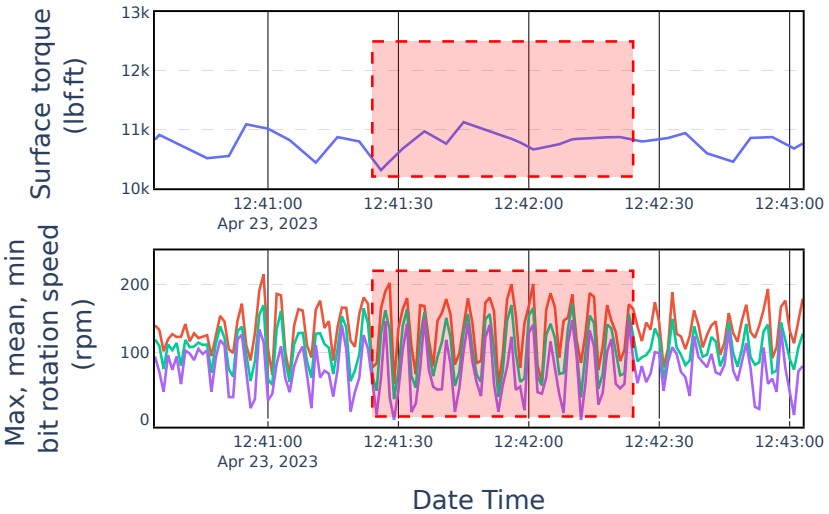

Figure 13: Variation in surface torque (measured at the surface) and bit rotation speed (measured downhole) during a test sequence. Stick-slip behavior occurs at the bit, but no surface torque variation is observed during propagation (due to friction). The predicted SSI using the trained ADG model for the 60 second red sequence is 0.14 and the true SSI is 0.87.

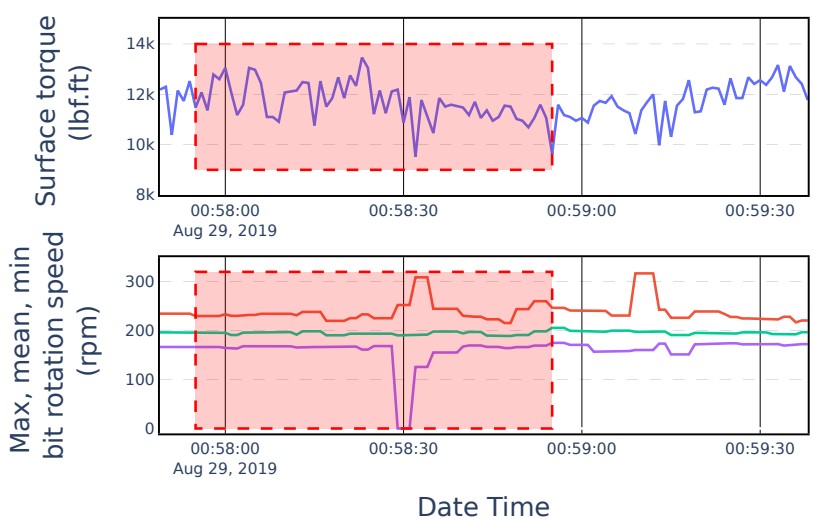

Figure 14: Variation in surface torque (measured at the surface) and bit rotation speed (measured downhole) during a test sequence. In this sequence, a peak in downhole bit rotation speed during the 60 second red sequence generates a high true SSI (0.72), despite no significant variation in surface measurements, resulting in a low predicted SSI (0.21).

## 6 Conclusion

In this paper, we explored the application of domain generalization techniques to time series data for predicting the Stick-Slip Index (SSI) in drilling operations. Our approach used 60 second sequences of 1 Hz surface drilling data as inputs. We compared the performance of the Adversarial Domain Generalization (ADG) model with the Invariant Risk Minimization (IRM) model, a traditional baseline model, and a TL approach.

To optimize key parameters including the regularization coefficient, the architecture of the model, and the weighting coefficient for the ADG and the IRM models, we employed a grid search methodology. Our results demonstrate that ADG and IRM models significantly enhance the generalization capabilities of the SSI prediction model, achieving improvements of 10% and 8% over the baseline model, respectively. Notably, the ADG model slightly outperforms the IRM model. Additionally, the implementation of TL on a pretrained model (whether ADG or baseline) demonstrated improved performance. Even after TL application, the ADG model outperforms the baseline model, which highlights the generator's ability to map the training data sequences into a space where the SSI predictor cannot differentiate among them.

The proposed model can readily be used to detect stick-slip on new wells with no need for downhole sensors to be installed. In particular, severe events are detected 60% of the time. This constitutes a new tool for drilling operators to monitor drilling vibrations in real-time, and reduce drilling costs. In future works, we intend to implement ADG using a larger variety of training wells. Additionally, we plan to explore the Adversarial Domain Adaptation (ADA) technique and evaluate its performance in comparison to TL.

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
