# OpenReview forum: "Domain Generalization for Time Series: Enhancing Drilling Regression Models for Stick-Slip Index Prediction"
_TMLR — Accepted by TMLR_

### Review · Reviewer_QDMe · 2024-12-13

**Summary Of Contributions:**

In this paper, a new domain generalization method is proposed for predicting the stuck slip index (SSI) during drilling. SSI is a key index used to measure the strength of torsional vibration under the bit, and the stuck slip phenomenon in torsional vibration usually affects drilling efficiency and may even lead to equipment damage

**Audience:**

Yes

**Claims And Evidence:**

Yes

**Requested Changes:**

See weakness

**Strengths And Weaknesses:**

The submission introduces a novel domain generalization approach for predicting the Stick-Slip Index (SSI) using adversarial domain generalization. This method effectively addresses the challenge of generalizing across wells with distinct characteristics, which is often a major limitation in traditional machine learning models.

The submission has significant practical applications in the field of drilling and oil/gas exploration, where accurate prediction of stick-slip events is critical for optimizing drilling efficiency and preventing equipment damage.


The authors perform a grid search for hyperparameter tuning, ensuring that the model is optimized for performance. This level of detail in model selection improves the robustness of the results and contributes to the reliability of the proposed approach.

----

weakness


The paper does not present significant technical innovation. The use of adversarial learning and other techniques (such as domain generalization) are well-established methods already widely applied in Out-of-Distribution (OOD) problems. While these methods are certainly valuable, the paper does not demonstrate a novel approach or an innovative contribution that justifies its publication in TMLR, which typically focuses on papers offering new theoretical insights or novel algorithmic advancements.
Insufficient Error Analysis:

The paper does not provide a detailed error analysis for the model's predictions. While performance metrics such as MSE and DTW are reported, there is a lack of thorough investigation into why the model fails in certain cases or how errors are distributed across the test set. An in-depth analysis of failure cases, such as data-specific errors, model limitations, or domain shifts, would provide valuable insights into how the model can be improved and in which scenarios it is less effective.
Introduction Section Needs Improvement:

The Introduction section is overly long and seems to blend the related work with the contribution of the paper, leading to a lack of focus. It feels more like a general literature review rather than a clear statement of the problem and novel contributions of the paper. The information density is low, and the key findings and novelty of the work are not highlighted effectively. A more concise and focused introduction would help set the context more clearly and emphasize the core research question and contribution.
Potential Improvements with Stronger Adversarial Learning Strategies:

If the proposed adversarial learning approach is indeed effective, it would be useful to explore whether stronger adversarial techniques (such as Invariant Risk Minimization (IRM) or ELS [Free Lunch for Domain Adversarial Training: Environment Label Smoothing] could provide further performance improvements. While the current method shows some promise, incorporating or comparing with these more advanced techniques could enhance the effectiveness and robustness of the model.


It is noted that the reviewer has limited familiarity with the application area (e.g., drilling operations), which restricts the ability to assess the domain-specific contributions of the paper.

---

> ### Author Response · Authors · 2025-01-17
>
> Thank you for your valuable feedback and insightful suggestions, which greatly improved our manuscript.
>
> Weakness:
>
> 1 - "The paper does not present significant technical innovation. The use of adversarial learning and other techniques (such as domain generalization) are well-established methods already widely applied in Out-of-Distribution (OOD) problems. While these methods are certainly valuable, the paper does not demonstrate a novel approach or an innovative contribution that justifies its publication in TMLR, which typically focuses on papers offering new theoretical insights or novel algorithmic advancements."
>
>
> We agree that our article is primarily oriented toward the application of domain generalization techniques to drilling applications. TMLR invites the submission of papers accounting for applications of existing techniques that shed light on the strengths and weaknesses of the methods, which is the criterion on which we based our submission.
>
> The industrial data we used to apply domain generalization methods present particularly challenging processing issues. Nonetheless, it is noteworthy that domain generalization methods lead to significant improvements in model quality. Following your suggestions, we have added several sections to the article. These include a discussion of the conditions under which the domain generalization techniques were applied (Section 3.4), the limitations of the domain generalization approach - which, by itself, cannot address all the challenges posed by the data - and a comparison between the adversarial domain generalization approach and invariant risk minimization. We thank you for these suggestions, which enhance the quality of the paper and allow for a deeper analysis of the improvements brought by domain generalization methods on the analysis of real industrial data.
>
>
>
> 2 - "Insufficient Error Analysis: The paper does not provide a detailed error analysis for the model's predictions. While performance metrics such as MSE and DTW are reported, there is a lack of thorough investigation into why the model fails in certain cases or how errors are distributed across the test set. An in-depth analysis of failure cases, such as data-specific errors, model limitations, or domain shifts, would provide valuable insights into how the model can be improved and in which scenarios it is less effective. "
>
>
>
> Thank you for pointing this out. As recommended by the reviewer, additional error analysis of the model's predictions has been included at the end of Section 5.3.1, where confusion matrices of the predicted SSI are presented. Furthermore, a new section (Section 5.3.3) has been added at the end of the paper to explore the reasons behind the model's mispredictions. These reasons include issues during the data processing phase, such as synchronization problems between measured surface and downhole data, leading to mislabeled sequences. Another challenge is unobservable stick-slip events, particularly in lateral wells, where downhole vibrations are attenuated during propagation to the surface due to friction and viscous damping, making them undetectable at low frequencies. Additionally, some downhole sudden variations (peaks) may yield high SSI values without indicating actual stick-slip occurrences. On the model training side, domain mismatch poses a significant challenge. This occurs when the model is trained on insufficient data that fails to adequately represent the target domain, limiting its ability to generalize effectively. Those four distinct reasons are discussed, accompanied by examples for clarification (please refer to the paper for more details).

---

> > ### Author Response · Authors · 2025-01-17
> >
> > 3 - "Introduction Section Needs Improvement: The Introduction section is overly long and seems to blend the related work with the contribution of the paper, leading to a lack of focus. It feels more like a general literature review rather than a clear statement of the problem and novel contributions of the paper. The information density is low, and the key findings and novelty of the work are not highlighted effectively. A more concise and focused introduction would help set the context more clearly and emphasize the core research question and contribution. "
> >
> >
> >
> > Thank you for this remark. As suggested, we have revised the introduction. It now begins by discussing the generalization problem in machine learning models broadly, narrowing down to its relevance in the drilling field and, more specifically, in stick-slip detection. We provide a definition of the stick-slip phenomenon in drilling, followed by a brief overview of existing solutions developed to detect it and their limitations, avoiding excessive drilling-specific details since this is not a drilling-focused journal.
> >
> > The introduction also covers data-driven models developed for stick-slip detection and their challenges with generalization. Proposed solutions to address the generalization problem in SSI detection are discussed, including approaches beyond drilling applications, such as domain adaptation and its applications. We then highlight the limitations of domain adaptation, leading to the motivation for exploring domain generalization. Finally, we outline the paper's objective: to apply domain generalization techniques for time series data in the context of SSI prediction.
> >
> >
> >
> >
> >
> >
> >
> > 4 - "Potential Improvements with Stronger Adversarial Learning Strategies: If the proposed adversarial learning approach is indeed effective, it would be useful to explore whether stronger adversarial techniques (such as Invariant Risk Minimization (IRM) or ELS [Free Lunch for Domain Adversarial Training: Environment Label Smoothing] could provide further performance improvements. While the current method shows some promise, incorporating or comparing with these more advanced techniques could enhance the effectiveness and robustness of the model. "
> >
> >
> >
> > Thank you for pointing this out. As you suggested, we included another comparison using a different technique, Invariant Risk Minimization (IRM). In the revised version of the paper, we train an IRM model and compare its performance with that of the Adversarial Domain Generalization model and the baseline model. Both ADG and IRM models significantly enhance the generalization capabilities of the SSI prediction model, achieving improvements of 10% and 8% over the baseline model, respectively. All results, as well as details about the training process and IRM architecture, are now included in the paper.

---

### Review · Reviewer_ChHd · 2025-01-09

**Summary Of Contributions:**

Within the drilling industry, detection of stick-slip is of particular interest as it can significantly impact drilling performance and equipment integrity. Furthermore, the challenge of transfer learning to the target domain and training on diverse source domains with potentially disparate underlying distributions is present in the context of data gathered from different drilling wells.

To this end, this paper adapts the Domain-Adversarial Network (DAN) proposed in Ganin et al. (2016), using surface measurements to predict the stick-slip index (SSI), which is a metric quantifying the severity of a stick-slip. In particular, DAN aims to learn domain-invariant representations of surface measurements across different source distributions, i.e. data from different wells. This is realized by maximizing the discriminator's loss in the typical adversarial setting involving a generator of the representation of the input data and discriminator serving as proxy of how distinguishable different representations generated are. To clarify, the DAN considered in this paper consists of the following three networks:

- Generator: Alternating LSTM cells with Layer Normalization (LN) layers, taking in surface measurements and outputs its representation.
- Discriminator: Fully Connected Network (FCN), taking in representation and predicts the well the representation is originating from.
- Predictor: FCN, taking in representation and predicts the corresponding SSI.

This model is compared against the baseline model from Yahia et al. (2024a), which the authors claim to have the same generator architecture but different predictor, although this predictor has identical loss function, metrics, and optimizer as that of DAN. I cannot verify the correctness regarding the statements about the baseline model due to the paywall present in accessing Yahia et al. (2024a) as it is published in a rather niche proceeding. See for questions regarding this in **Requested Changes**.

After grid search for the hyperparameters of DAN, DAN appears to outperform the baseline model by around 10% on all test wells. Similarly, after fine-tuning the models on the target distribution, DAN still outperforms the baseline model.

**Audience:**

Yes

**Broader Impact Concerns:**

None.

**Claims And Evidence:**

Yes

**Requested Changes:**

**Major**
- I believe that it is essential to justify the choice of the baseline model via literature study. This is because if the baseline model chosen is deprecated for today's practice then comparison to it as evidence of superiority of proposed model is meaningless. Therefore, I would like to request incorporation of (1) **what are the alternatives to this choice as the baseline model** and (2) **why this particular model makes a good choice as the baseline model**.
- I believe that it is essential to discuss the applicability of the assumptions mentioned for DAN to the surface measurement data in relation to SSI. Model reliability is essential and verifying its assumptions builds trust in its use in reality. I understand that this may be difficult without comprehensive analysis but in my opinion **it cannot be left blank** if the assumptions are mentioned in the first place.
- Recall that I do not have access to Yahia et al. (2024a) in which the baseline model used in this paper is proposed as mentioned in **Summary Of Contributions**. For the baseline model, what is exactly its architecture? Is it also DAN? Furthermore, in section 4.2, it is mentioned:
*"The only addition is a final dense layer with a single neuron responsible for outputting the predicted SSI."*
. However, for DAN, it is mentioned that *"The SSI-predictor has a single output neuron responsible for predicting the SSI."*. To me, this means that the predictor for the baseline model is exactly the same as that of DAN. **My request is therefore to clearly state the difference between the two.**

**Minor**
- I am unsure about the second equality of Equation 2 on the expected risk. According to the change of variables, for the second equality to hold, $R'(x)P(R(x))=P(x)$, where $P$ is the probability measure on $D_T$. **This should be justified or mentioned** in the text unless I am miscalculating something.

**Inessential suggestion**
- Personally, I am not a big fan of this idea of making the representation agnostic to the source distribution via a discriminator. It is just unreliable to what extent the generator is able to identify the separating set in the causal graph or what one might call the invariant features. For this reason, I am curious if the authors have encountered any difficulty in training the DAN that reflects this problem.

**Strengths And Weaknesses:**

**Strengths**
- The paper is for the most part well-written in terms of readability, structure and presentation.
- Assumptions under which DAN is effective is clearly described.
- DAN outperforms the baseline model.
- Data with similar characteristic are consistently used either training or test data, but not both, making efficacy of DAN more credible in the context of transfer learning.

**Weaknesses**
- Choosing the model from Yahia et al. (2024a) as the baseline is not justified based on literature review, i.e. (1) what are the alternatives to this choice as the baseline model and (2) why this particular model makes a good choice as the baseline model.
- Missing analysis on to what extent the assumptions of DAN hold in this case.

---

> ### Author Response · Authors · 2025-01-17
>
> Thank you for your valuable feedback and insightful suggestions, which greatly improved our manuscript.
>
> Weaknesses
>
> 1 - "Choosing the model from Yahia et al. (2024a) as the baseline is not justified based on literature review, i.e. (1) what are the alternatives to this choice as the baseline model and (2) why this particular model makes a good choice as the baseline model. "
>
> Thank you for your feedback. Overall, the literature dedicated to the prediction of downhole vibrations remains relatively scarce to date. Therefore, there is no consensus on the optimal architecture for this problem. Among the recent papers on this topic, Saadeldin et al., 2023 use relatively simple ML techniques including support vector machines (SVM), radial basis functions or functional networks to estimate downhole vibrations. Similarly, the recent paper Elahifar et al. 2024 relies on decision trees to perform the stick-slip prediction. To the best of our knowledge, our approach is therefore one of the first to use deep learning techniques to perform SSI prediction.
>
> To select an architecture, we tried different approaches for the baseline model, including recurrent neural networks, transformers and LSTM. In the end, we kept the LSTM architecture, which yielded the best results in terms of accuracy and in terms of execution time when compared to the transformers architecture. This comparison is mentioned in the paper (page 8, end of 4.1):
>
>
> In recent years, the transformer model (Vaswani, 2017) has gained popularity over 	LSTM, as it eliminates the sequential processing of LSTMs by using self-attention mechanisms to process all 	time steps in parallel. This parallelization is often seen as advantageous for time series tasks. However, in choosing the architecture for the generator, we compared LSTMs architecture with transformers. During testing, the transformer model required significantly more time to train (training time increased by a factor of five), and its performance was less favorable compared to LSTMs, which are more efficient with smaller datasets. For these reasons, we opted to use LSTM layers for the generator architecture.
>
>
> 2 - "Missing analysis on to what extent the assumptions of DAN hold in this case. "
>
> Thank you for highlighting this. As mentioned in the paper, domain generalization techniques are theoretically justified under the following assumptions:
>
> Covariate shift: This assumption requires the target function to remain consistent across all domains. In other words, even if the distribution of the input features changes between the source domain and the target domain, the conditional distribution of the labels given the features remains the same. In our case, the underlying relationship between the input surface measurements and SSI values is expected to stay stable, despite potential variations in environmental conditions (domains). For instance, drilling in sandstone may result in higher torque variability compared to shale, but the fundamental process of stick-slip occurrence and its prediction from surface measurements are governed by consistent physical principles, ensuring the target function's stability across domains. However, as stated in the paper, according to (David et al., 2010), only the next two assumptions are necessary for applying model generalization. Moreover, the covariate shift assumption is considered less restrictive, as it can be validated independently of the feasibility of model generalization.
>
>
>
> Similarity between features distributions across domains: In our case, there is always a degree of similarity between feature distributions across domains, even when the input feature distributions vary. This is because drilling operations typically rely on standardized tools and techniques, which help preserve consistency in surface measurement patterns across wells. As a result, the overall structure of input feature distributions remains comparable across domains.
>
>
> Existence of a suitable mapping function: Prior studies and data analysis have demonstrated a correlation between surface measurements, particularly surface torque, and SSI, supporting the assumption that a suitable mapping function exists. Machine learning models have proven effective in capturing these relationships in similar applications. For instance, our work in Yahia et al. (2024a) showcases the feasibility of developing a mapping function for SSI prediction. Even if the mapping function is not perfect, a regression model can still approximate it with a reasonable degree of accuracy.

---

> > ### Author Response · Authors · 2025-01-17
> >
> > Requested Changes:
> >
> > Major:
> >
> > 1 - "I believe that it is essential to justify the choice of the baseline model via literature study. This is because if the baseline model chosen is deprecated for today's practice then comparison to it as evidence of superiority of proposed model is meaningless. Therefore, I would like to request incorporation of (1) what are the alternatives to this choice as the baseline model and (2) why this particular model makes a good choice as the baseline model."
> >
> > As suggested by the reviewer, additional details regarding the selection of the baseline model have been incorporated into the paper. For more information, please refer to subsection 4.1.
> >
> > 2 - "I believe that it is essential to discuss the applicability of the assumptions mentioned for DAN to the surface measurement data in relation to SSI. Model reliability is essential and verifying its assumptions builds trust in its use in reality. I understand that this may be difficult without comprehensive analysis but in my opinion it cannot be left blank if the assumptions are mentioned in the first place. "
> >
> > As suggested by the reviewer, additional details regarding the validity of the assumptions for applying DG have been included. Please refer to Subsection 3.1.
> >
> >
> >
> > 3 - "Recall that I do not have access to Yahia et al. (2024a) in which the baseline model used in this paper is proposed as mentioned in Summary Of Contributions. For the baseline model, what is exactly its architecture? Is it also DAN? Furthermore, in section 4.2, it is mentioned: "The only addition is a final dense layer with a single neuron responsible for outputting the predicted SSI." . However, for DAN, it is mentioned that "The SSI-predictor has a single output neuron responsible for predicting the SSI.". To me, this means that the predictor for the baseline model is exactly the same as that of DAN. My request is therefore to clearly state the difference between the two."
> >
> > Thank you for pointing this out. As described in the paper, the baseline architecture comprises a series of Long Short-Term Memory (LSTM) layers combined with Layer Normalization (LN) layers, followed by an output dense layer with a single neuron responsible for predicting the Stick-Slip Index (SSI).
> >
> > In contrast, the DAN architecture is composed of three main components, as illustrated in Figure 3 of the paper: the generator, the SSI predictor, and the domain classifier. The generator, built using LSTM and LN layers, maps the sequences of surface measurements into a latent space. The SSI predictor and domain classifier then use the generator's output as input to predict the SSI and the environment, respectively. Both the SSI predictor and domain classifier are implemented using dense layers, with the SSI predictor featuring a single-neuron dense output layer.
> >
> > In response to the reviewer's suggestion, we have added further details about the baseline model architecture. Please refer to Subsection 4.1 for more information.
> >
> >
> >
> > Minor:
> >
> > 1 - "I am unsure about the second equality of Equation 2 on the expected risk. According to the change of variables, for the second equality to hold, R′(x)P(R(x))=P(x), where P is the probability measure on DT. This should be justified or mentioned in the text unless I am miscalculating something."
> >
> > In Equation (2), the expectations are not computed with the same distribution, but with respect to the original distribution of x for the left-hand side and to the image distribution of x by the map R for the right-hand side. The Jacobian R’ that you mention is therefore already incorporated in the definition of the image distribution.
> >
> >
> >
> > Inessential suggestion:
> >
> > 1 - "Personally, I am not a big fan of this idea of making the representation agnostic to the source distribution via a discriminator. It is just unreliable to what extent the generator is able to identify the separating set in the causal graph or what one might call the invariant features. For this reason, I am curious if the authors have encountered any difficulty in training the DAN that reflects this problem. "
> >
> >
> >
> > Thank you for this remark. The main challenge lies in analyzing the new projection of input sequences in the latent space. Specifically, for the domain classifier, by the end of training, it fails to accurately classify sequences into their respective environments. Instead, it predicts a uniform probability vector $$\{\frac{1}{N_i}, i\in  \llbracket1,M\rrbracket\}$$ where M s the number of test environments and Ni is the number of sequences per test environment i. This result confirms that the learned representation successfully removes domain-specific variations, achieving the intended behavior for domain generalization. However, verifying this outcome solely using the generator's outputs remains challenging.

---

### Review · Reviewer_9GhK · 2025-01-12

**Summary Of Contributions:**

The paper presents an empirical study on the effectiveness of domain generalization methods for predicting the Stick-Slip Index (SSI) in drilling applications. The authors compared one adversarial domain generalization approach to a standard baseline regression model, demonstrating that domain generalization techniques lead to a more robust overall predictor that generalizes well to prediction tasks from unseen wells. Additionally, the authors showed that incorporating few-shot fine-tuning during testing further enhances model performance, for both the baseline and DG, allowing the trained model to adapt effectively to the specific characteristics of data from unseen wells.

**Audience:**

No

**Claims And Evidence:**

Yes

**Requested Changes:**

Please kindly refer to the weaknesses above.

**Strengths And Weaknesses:**

## Strengths
- The paper is mostly clearly written and easy to follow. Sufficient context is provided for the drilling SSI prediction as a problem setup, particularly in explaining the significance of the SSI and the challenges associated with predicting it in drilling operations .

## Weaknesses
- The paper presents mostly empirical results. The findings are somewhat expected, and there are very limited new insights regarding the domain generalization methods in general or specifically in the context of SSI prediction applications. The authors primarily focus on the adversarial domain generalization technique -- the only DG technique being evaluated and compared, which may restrict the comprehensiveness of the empirical study.
- Only one 'traditional' domain generalization technique is being studied, which may limit the paper's depth and prevent the authors from making new discoveries. Perhaps, If more domain generalization techniques were examined, the authors could potentially conduct a comparison and evaluation of their effectiveness, leading to more interesting results or new insights.
- Were there any important modifications required for applying the adversarial domain generalization approach to the SSI prediction task? Perhaps the authors could highlight some challenges that demonstrate this is not merely a straightforward application of existing methods.

---

> ### Author Response · Authors · 2025-01-17
>
> Thank you for your valuable feedback and insightful suggestions, which greatly improved our manuscript.
>
> 1 - “The paper presents mostly empirical results. The findings are somewhat expected,and there are very limited new insights regarding the domain generalization methods in general or specifically in the context of SSI prediction applications. "
>
> Thank you for your feedback. The primary novelty of our work lies in applying domain generalization (DG) methods to a regression problem concerning time series data, an area where such techniques remain under-explored. Generally, DG methods are tested on classification problems with image data, and their application to time series remains limited. Furthermore, our model is tested on real drilling data, making this a real-world prediction problem that is particularly complex due to the significant generalization challenges between wells.
>
> In the specific context of SSI prediction, a well-known difficulty is that machine learning algorithms trained on a given well or on wells belonging to the same rigs generalize poorly when applied to new wells. To our knowledge, our paper is the first to explore the use of domain generalization (DG) to mitigate this essential issue. In addition, most published papers in this field focus on classification models, which classify sequences of surface measurements into classes based on different stick-slip severity levels, rather than on regression models aimed at predicting the actual SSI values.
>
> To provide additional insights on domain generalization methods in the context of SSI prediction, we included a detailed analysis in section 5.3.3 of the paper on the issues that hinder the ability of the ML models to generalize when applied to new drilling contexts. These factors include issues during the data processing phase, such as synchronization problems between measured surface and downhole data, leading to mislabeled sequences. Another challenge is unobservable stick-slip events, particularly in lateral wells, where downhole vibrations are attenuated during propagation to the surface due to friction and viscous damping, making them undetectable at low frequencies. Additionally, some downhole sudden variations (peaks) may yield high SSI values without indicating actual stick-slip occurrences. On the model training side, domain mismatch poses a significant challenge. This occurs when the model is trained on insufficient data that fails to adequately represent the target domain, limiting its ability to generalize effectively. Despite these difficulties, as demonstrated in Fig. 9, our approach proves particularly effective in identifying severe stick-slip events, the most critical scenario in drilling operations.
>
>
>
> 2 - "Only one 'traditional' domain generalization technique is being studied, which may limit the paper's depth and prevent the authors from making new discoveries. Perhaps, If more domain generalization techniques were examined, the authors could potentially conduct a comparison and evaluation of their effectiveness, leading to more interesting results or new insights."
>
>
>
> Thank you for highlighting this point. To address this issue, we included an analysis of the Invariant Risk Minimization (IRM) approach in the paper, which is another commonly used approach to perform Domain Generalization (DG). In the revised version, we train an IRM model and compare its performance with that of the Adversarial Domain Generalization (ADG) model and the baseline model. The ADG and IRM models significantly enhance the generalization capabilities of the SSI prediction models, achieving improvements of 10% and 8% over the baseline model, respectively. All results, as well as details about the training process and IRM architecture, are now included in the paper.

---

> ### Author Response · Authors · 2025-01-17
>
> 3 - "Were there any important modifications required for applying the adversarial domain generalization approach to the SSI prediction task? Perhaps the authors could highlight some challenges that demonstrate this is not merely a straightforward application of existing methods. "
>
> Thank you for pointing this out. Here are the main challenges that we faced for training the model:
>
>        1 - Selecting a model architecture specifically adapted to time series data: as noted in the last paragraph of Section 4.1, we evaluated several architectures, including transformers. We noticed that transformers required substantially more training time (a fivefold increase) and underperformed compared to LSTMs, which proved more efficient and better suited for smaller datasets. This is the main reason why we ultimately chose LSTMs for this study.
>
>        2 - Hyperparameter tuning and domain imbalance: our approach involved multiple hyperparameters, such as the regularization coefficient, the number of hidden layers in the generator, and the weighting coefficient. Additional parameters, like the architecture and latent space size for the generator, domain classifier, and SSI predictor, further complicated optimization. Given limited computational resources, we fixed some parameters and applied grid search to others, which was a time-intensive and nontrivial process. Domain imbalance compounded these challenges, as training data was unevenly distributed across domains.
>
>       3 - Training process: the training process itself required careful management due to the complexity of the adversarial model. Simultaneously training three components with distinct roles necessitated precise tuning of batch size, learning rate, number of epochs, and synchronization of training stages to achieve optimal performance
>
>     4 - Performances evaluation: evaluating model performance was particularly challenging with time series data. Comparing two time series with variable time shifts is not straightforward. One potential solution is the use of Dynamic Time Warping (DTW), a computational technique for measuring the similarity between two time series that may differ in timing or speed. We applied DTW to assess the similarity between the actual and predicted SSI validation data. However, a drawback of this method is that it is not easily interpretable.

---

### Decision · Action_Editor_iF2E · 2025-02-18

**Recommendation:** Accept as is

**Comment:**

This paper evaluates the performance of domain generalization techniques, specifically Adversarial Domain Generalization (ADG) and Invariant Risk Minimization (IRM), for predicting the Stick-Slip Index (SSI) in drilling operations, analyzing their suitability for time series regression tasks with limited data.

The AE and reviewers acknowledged the paper's practical value. Given the empirical focus of this research, the authors are encouraged to make their code publicly available to support practical implementations.

**Audience:**

Yes

**Claims And Evidence:**

Yes.